# Comparative connectomics of dauer reveals developmental plasticity

Hyunsoo Yim[1,7], Daniel T. Choe[1,7], J. Alexander Bae[2,7], Myung-kyu Choi[2], Hae-Mook Kang[2], Ken C. Q. Nguyen [3], Soungyub Ahn[1], Sang-kyu Bahn[4,6], Heeseung Yang[1], David H. Hall[3], Jinseop S. Kim [4,5] ✉ & Junho Lee [1,2] ✉

A fundamental question in neurodevelopmental biology is how flexibly the nervous system changes during development. To address this, we reconstructed the chemical connectome of dauer, an alternative developmental stage of nematodes with distinct behavioral characteristics, by volumetric reconstruction and automated synapse detection using deep learning. With the basic architecture of the nervous system preserved, structural changes in neurons, large or small, were closely associated with connectivity changes, which in turn evoked dauer-specific behaviors such as nictation. Graph theoretical analyses revealed significant dauer-specific rewiring of sensory neuron connectivity and increased clustering within motor neurons in the dauer connectome. We suggest that the nervous system in the nematode has evolved to respond to harsh environments by developing a quantitatively and qualitatively differentiated connectome.

A fundamental question in neurobiology is how the nervous system governs behavior[1]. Specifically, researchers seek to understand how an animal's nervous system assimilates diverse sensory information from its external surroundings to make adaptive decisions for survival. Elucidating and analyzing connectomes of animals is one way to address this question. A more specific question in neurobiology in terms of development concerns the capacity of an animal to dynamically shape its nervous system throughout its development. In other words, how extensively does an animal modify its neuronal connections in response to physiological and external cues at different developmental stages? Comparative connectomics provides excellent opportunities to answer this question of developmental plasticity.

Since Sydney Brenner initiated his ambitious research on the connectome, the nematode *C. elegans* has been at the center of connectomics studies[1,2]. Connectomics studies using *C. elegans* have been successfully extended to comparative studies, comparing the adult nervous systems of both sexes[3,4], and investigating the neural development along the series of life stages[5]. Along the same lines of research

directions, but more focused on developmental plasticity, the study of the connectome of dauer, an alternative developmental stage, is particularly intriguing. Under unfavorable conditions, *C. elegans* enters the diapause state of dauer, and systematic anatomical changes occur to many tissues, acquiring physiological robustness and displaying various unique behavioral repertoires, such as being motionless, stopping pharyngeal pumping, and changing $Na^+$ and $CO_2$ attractiveness. (Fig. 1a[6–11]). Dauers also show a stage-specific hitchhiking behavior, nictation, in which dauers stand with their tails and wave the bodies to facilitate phoresy to other carrier animals. We previously showed that IL2 neurons are responsible for the nictation behavior, but it is not well understood why only dauers, but not other stage animals, are capable of the nictation behavior[12]. The remodeling of the connectome underlying these behavioral changes in dauer could provide insights on both the neural substrates of behaviors and the principles of the plasticity of animal development[13,14].

In this study, we report the complete connectome of a hermaphrodite dauer nerve ring (Fig. 1a), in comparison with the publicly

[1]Department of Biological Sciences, Seoul National University, Seoul 08826, South Korea. [2]Research Institute of Basic Sciences, Seoul National University, Seoul 08826, South Korea. [3]Dominick P. Purpura Department of Neuroscience, Albert Einstein College of Medicine, Bronx, NY 10461, USA. [4]Neural Circuits Research Group, Korea Brain Research Institute, Daegu 41062, South Korea. [5]Department of Biological Sciences, Sungkyunkwan University, Suwon-si, Gyeonggi-do 16419, South Korea. [6]Present address: Cognitive Science Research Group, Korea Brain Research Institute, Daegu 41062, South Korea. [7]These authors contributed equally: Hyunsoo Yim, Daniel T. Choe, J. Alexander Bae. ✉e-mail: jinseopskim@skku.edu; elegans@snu.ac.kr

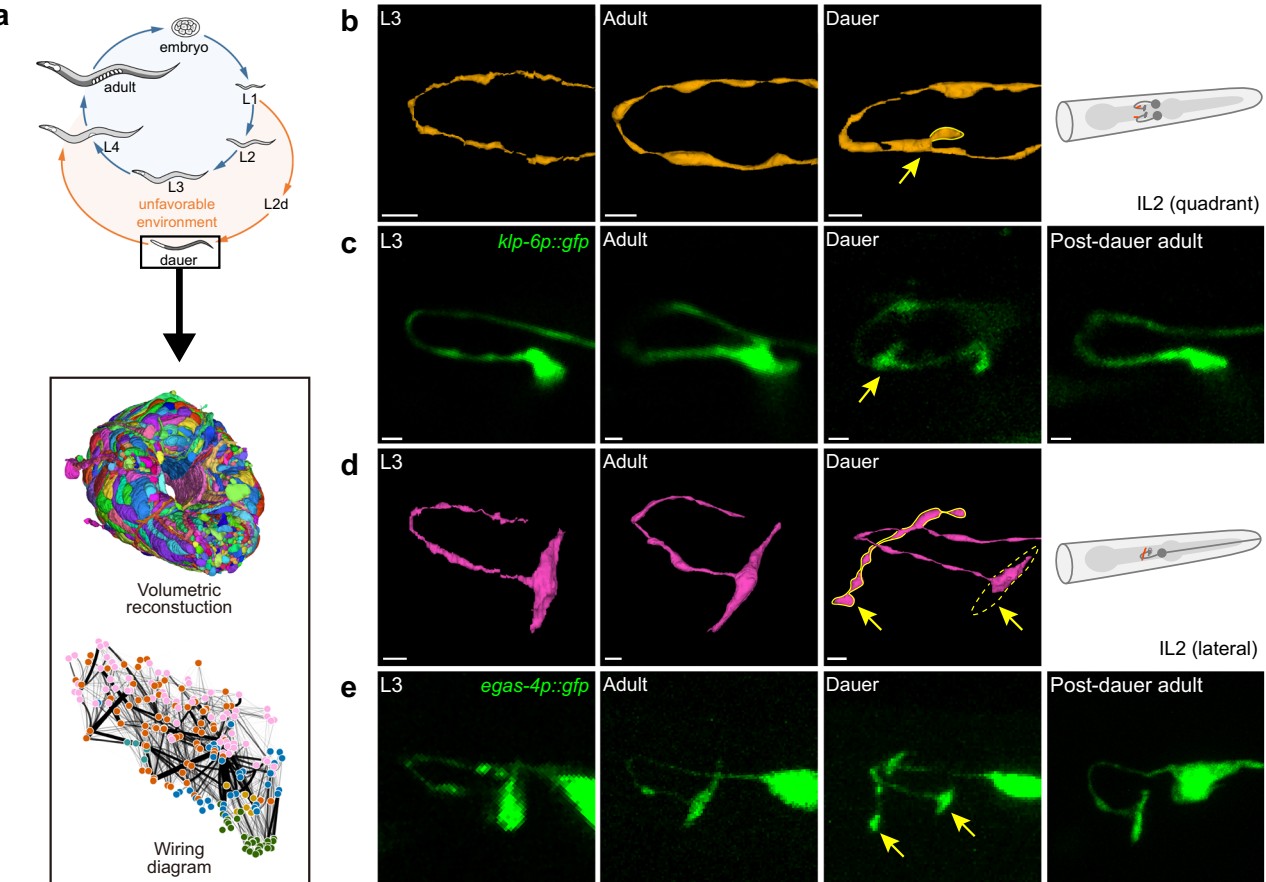

**Fig. 1 | Volumetric reconstruction of the dauer nerve ring. a** Volumetric reconstruction and wiring diagram (pink: sensory neuron, orange: interneuron, blue: motor neuron, green: body wall muscle) of dauer nerve ring. Volumetric reconstruction is colored to distinguish individual neurons. Arrangement of nodes in the wiring diagram is adopted from Witvliet et al.[5]. **b** Volumetric reconstruction of IL2DR (quadrant) neurons in L3, adult-2, dauer (from left) and schematic view of neuron in dauer stage (right). New protrusion emerges from the axon (yellow arrow, solid yellow). **c** Fluorescence images of IL2 quadrant neurons in L3, adult, dauer, and post-dauer adult (from left). IL2 lateral neurons were marked with GFP using the *klp-6* promoter. **d** Volumetric reconstruction of IL2R (lateral) neurons in L3, adult-2, dauer (from left) and schematic view of neuron in dauer stage (right). New branches emerge from the axon (yellow arrow, solid yellow), and the terminal swelling shrinks (yellow arrow, dashed yellow). **e** Fluorescence images of IL2 lateral neurons in L3, adult, dauer, and post-dauer adult (from left). IL2 lateral neurons were marked with GFP using *egas-4* promoter. **b–e** Scale bars: 1 μm. **b, d** Adapted from WormAtlas, Altun, Z.F., Herndon, L.A., Wolkow, C.A., Crocker, C., Lints, R. and Hall, D.H. (ed.s) 2002-2024.

available 10 connectome datasets consisting of L1 - L4 and adult stages from previous studies (Supplementary Fig. 1a[2,4,5,15,16]). We show that local and modest, but significant, morphological changes of dauer neuronal arbors accompany synaptic changes, while most synapses remain intact through development. We also report the results of the graph theoretical analyses of the dauer connectome for the purpose of identifying the structural changes in the network topology that affect the dynamics of the overall neural network. Our work implies that substantial, reversible changes in behavior can be accomplished within a static set of neurons, through plastic alterations among their local synaptic connections.

## Results

### Volumetric reconstruction of dauer nerve ring reveals morphological changes in neurons

A *C. elegans* dauer nerve ring sample was sectioned and imaged at every 50 nm, resulting in 364 serial transmission electron microscopy (TEM) images. The dense volumetric reconstruction (Fig. 1a, Supplementary Fig. 1b, Methods) yielded the processes of 181 neurons in the nerve ring, whose number is only modestly different from those of the other late developmental stages (Supplementary Fig. 1a). After the volumetric reconstruction and cell identification, we visually inspected the neural processes using the 3D renderings (Supplementary Data 1).

We noticed the changes in dauer with varying magnitudes and forms, the most prominent of which were categorized into Branching (B), small Protrusion (P), Retraction (R), and Shrinkage (S) of the neurites and terminal swellings. Unlike the minor anatomical changes except for growth in size observed across other development stages[5], the changes in dauer were often more dramatic as shown in Supplementary Fig. 2.

The IL2 neurons are a family of six sensory neurons, critical to the dauer-specific phoretic behavior called nictation; two of these are lateral cells and the rest are quadrant cells (dorsal and ventral)[2,12]. Dauer IL2 quadrant neurons add an extra protrusion at the axon bend (Fig. 1b, c). The dendritic arborizations of IL2 quadrant neurons were well-reported, but any structural change in axon was not reported before[17]. Thanks to the finer resolution of TEM, we discovered that remodeling occurs in axons of IL2 quadrant neurons as well. As a validation, we employed fluorescence microscopy to observe the IL2 quadrant neurons in different stages including a post-dauer adult (Fig. 1c), confirming that the extra protrusion of IL2 quadrant neurons appear only in dauer.

Dauer IL2 lateral neurons also show distinctive structural changes in dauer, adding an extra branch that emerges perpendicularly from the apex of an axon bend (Fig. 1d, e). We performed fluorescence quantification on IL2 lateral neurons selected due to their pronounced

morphological change easily observable with fluorescence. The branching of IL2 lateral neurons was checked for L4, dauer, and post-dauer L4, with $n = 20$ for each stage. We found that 0/20 of L4, 20/20 of dauer, and 1/20 of post-dauer L4 showed branching, indicating that dauer-specific morphological change is a very strong binary trait. In addition, the terminal swellings of IL2 lateral neurons were shrunken in dauer, losing contact with the IL2 dorsal neurons (Fig. 1d, e).

### Dauer connectome exhibits similarities and differences with other stages

In order to see how synaptic connections change according to the morphological changes, we analyzed the full nerve ring connectome of the dauer. We annotated the chemical synapses in the image stack, yielding a complete connectivity map between the reconstructed neurons (Fig. 1a[5,9]). A convolutional neural network (CNN) was trained to predict the locations of active zones (Methods) and the predicted active zones were segmented to distinguish each chemical synapse. The postsynaptic partners of an active zone, together forming synapses, were assigned by simulating neurotransmitter diffusion to the closest neighboring axons (Methods[5]). To ensure high accuracy, the detected synapses were proofread by human experts (Supplementary Fig. 1b–d and Methods).

Altogether, 2813 active zones and 6371 synapses were found in the dauer nerve ring (Supplementary Fig. 3a, b, Supplementary Data 2). Each active zone pairs with 2.26 postsynaptic partners on average (SD = 0.93; Supplementary Fig. 3c). 19% of the active zones form monadic and 81% polyadic synapses. The 6371 synapses form 2200 connections, meaning that a pair of pre- and postsynaptic neurons are connected by 2.90 synapses on average (Supplementary Fig. 3d). As the size of the active zone could be a proxy for the functional strength of the synapse, the sum of active zone sizes is regarded as the connection weight or strength between two neurons (Methods). The left-right homologous neuron pairs in the dauer show strong correlation, and homologous connection shows similar connection similarity as that of adults (Supplementary Fig. 3e, f, Methods). In summary, the dauer connectome is represented by a directed weighted graph with 181 nodes of neurons and 1995 neuronal connections, or in total 2200 connections including the neuromuscular junctions (Fig. 2a, Supplementary Data 3).

We could now compare the connectivity of dauer with other stages from earlier studies (Supplementary Fig. 1a[2,4,5,15,16]). We found that the total number of connections in the dauer nerve ring is similar to that of the adult nerve ring, thus deviating from the trend of linear gains with increasing larval ages (Fig. 2a[5]). Comparisons across all developmental stages, including dauer, show 317 connections in common, which we call the "stage-conserved" connections. Another 499 connections exist only in the dauer, which we call the "dauer-specific" connections. On the other hand, 31 connections exist only in other stages but not in dauer and we call these "dauer-loss" connections (Fig. 2c and Supplementary Data 4). Interestingly, dauer has over 25% of stage-specific connections, in contrast to the cases of other stages which have roughly 9% of stage-specific connections on average (Fig. 2c and Supplementary Fig. 3g). There are also many connections that are shared with other stages but show different connection weights in dauer (Fig. 2d). Wiring diagram differences of all individual neurons are summarized in Supplementary Data 5.

### Morphological changes and connectional changes are mutually associated

We examined whether the major morphological changes listed in Fig. 1 and Supplementary Fig. 2 were associated with the formation of new synapses (Supplementary Fig. 4). We found that all the morphological changes described in Supplementary Fig. 2 were accompanied with the formation of new synaptic inputs and outputs. For example, the IL2 quadrants (IL2Q) with new protrusions made new connections with various partners, among which the IL2Q → SAA was notable (Supplementary Fig. 4a, b). The IL2 lateral (IL2la) neurons with extra arbors established new IL2la → AUA and DVC → IL2la synapses (Supplementary Fig. 4c, d). Likewise, all the morphological changes described in Supplementary Fig. 2 were accompanied with the formation of new synaptic inputs and outputs.

Since some synaptic changes could involve less obvious morphological changes, we then inspected whether the connections with significant changes in their weights are associated with any morphological changes. We found that, for most connection changes described in Fig. 2d, thickening of neural processes or broadening of local swellings contributed to the changes in contact area and resulted in the establishment of new synapses. The connection with the largest weight in all dauer connections was ASG → ASI connection, both of which are amphid sensory neurons known to be involved in the dauer-entry decision[18]. The ASI → ASG connection weight was also dramatically increased in dauer (Fig. 3c, Supplementary Data 4). ASG and ASI did not form any synapse with each other in any reproductive stage although they did form several local contacts (Fig. 3a). At the corresponding location in the dauer, a new swelling developed in ASI while ASG tightly adhered to the swelling, producing a larger contact with new large synapses (Fig. 3a, c). The total ASG → ASI contact area in the dauer increased by a factor of 2.54 compared to adult (Methods, Supplementary Data 6). ASG output connections exhibited large weight loss in dauer. The ASG → AIA connection stood out among the 31 dauer-loss connections, as it is a strong connection in other developmental stages which has totally disappeared in dauer (Fig. 3b, c and Supplementary Data 5). The size of the local swelling in AIA decreased and the contact area between AIA and ASG also decreased. Thus, the total contact area between ASG and AIA was reduced by a factor of 0.27 in dauer compared to adult (Supplementary Data 6). We have observed a similar correlation between connectivity and contact in IL1 → RIC connection (Fig. 3d, f) and RIC → AVA and AVB connections (Fig. 3e, f). In dauer, the total contact area between IL1 and RIC increased by a factor of 6.32 representing strengthened IL1 → RIC connection. Moreover, the total contact area of RIC and AVA decreased and that of RIC and AVB increased by a factor of 0.51 and 5.98, respectively, which were consistent with connectivity changes.

### Connectivity changes are correlated with unique behavior in dauer

Next, we asked whether the connectivity changes in dauer are associated with the unique behavior of dauer. We first focused on the IL2 neurons, which are known to play an essential role in nictation[12]. We found that IL2 quadrant neurons established strong connection to RIG neurons in dauer compared to other stages (Fig. 2d, Supplementary Fig. 4b). In dauer, each IL2 process forms a new swelling, forming new contact and large synapse with RIG neurons (Fig. 4a), and synapses were confirmed by GRASP (Fig. 4b)[19]. Therefore, we hypothesized that RIG could be potential neurons involved in the neural circuit regulating nictation, serving as downstream neurons of IL2 neurons. To test this, we performed behavioral tests with RIG-ablated worms, and found that RIG-ablated worms exhibit deficit in nictation behavior (Fig. 4c). To observe the RIG neuronal effect, we performed behavioral tests with dauers bearing mutations in two genes, *cha-1* and *daf-10*, which are known to show poor nictation[12]. The nictation ratio of *cha-1* and *daf-10* mutant dauers was significantly increased by optogenetic activation of RIG (Fig. 4d, e), suggesting that RIG activation is sufficient for the nictation of dauer even in the absence of ciliated, functional IL2 neurons and that RIG neurons act downstream of IL2 in nictation. Consequently, strengthened anatomical IL2 → RIG connection in the dauer stage is believed to have a functional role leading to behavior.

Another candidate neurons for dauer-specific physiology and behavior were the ASG neurons as the connections of ASG showed the most dramatic change in strength (Fig. 3a). We found that ASG-ablated

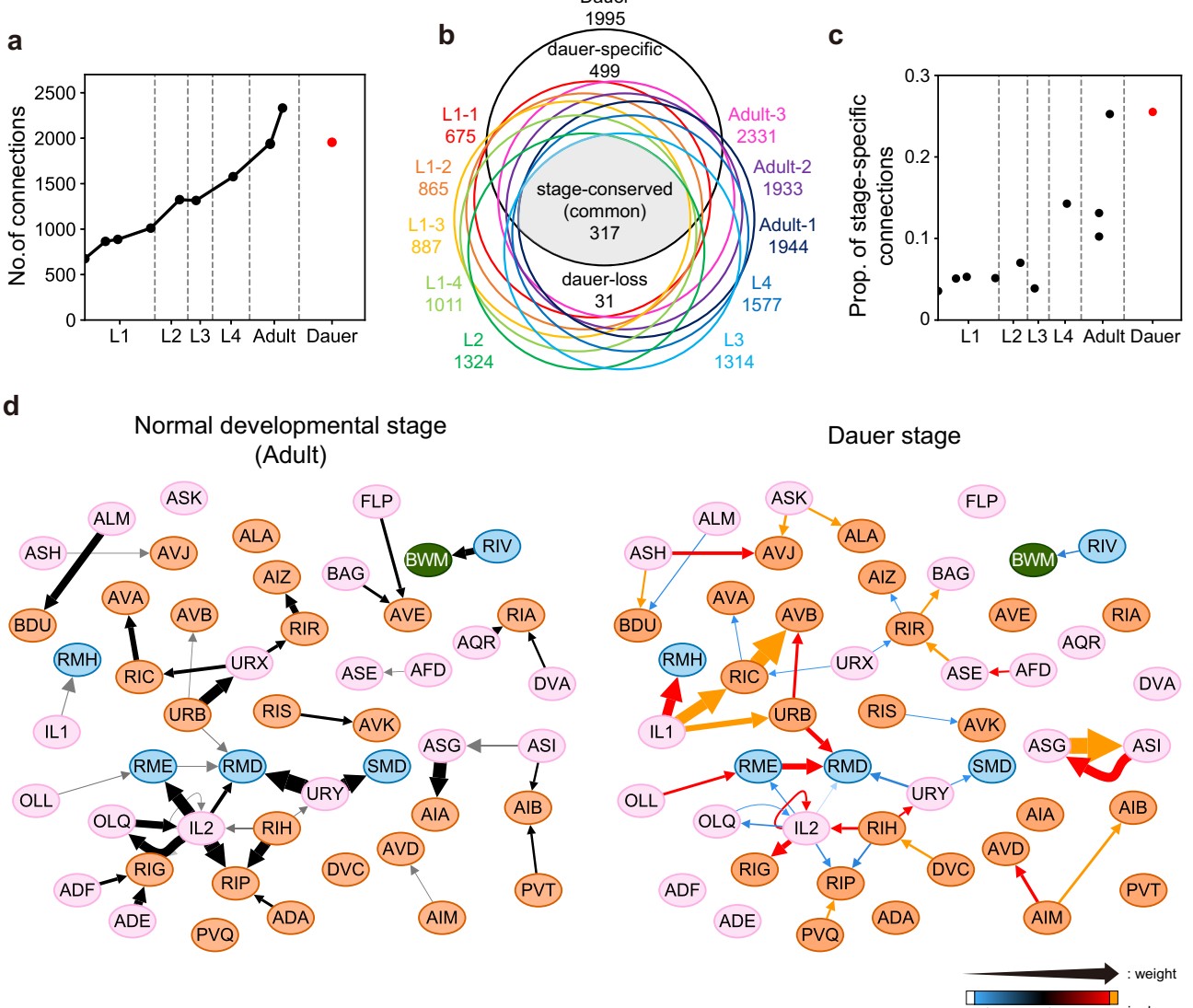

**Fig. 2 | Completion of dauer connectome. a** Total number of connections between neurons in different developmental stages. Total number of connections in dauer is similar to that of adults. **b** Venn diagram of connections between neurons. 499 connections are connections that only exist in dauer. Area of each region is not scaled to the number of connections. **c** Proportion of stage-specific connections within each dataset. Over a quarter of connections are dauer-specific connections, which are considerably higher than other stages. **d** Wiring diagram that includes representative changes in dauer connections (right) and wiring diagram of adult-2 stage (left) for the same set of neurons (pink: sensory neuron, orange: interneuron, blue: motor neuron, green: body wall muscle). The width of arrow indicates the relative connection weight and colors indicate changes in dauer relative to adult stage. Dauer-specific connections are represented with orange arrows. Connections with increased or decreased weight in dauer are represented with red or blue arrow, respectively.

worms exhibited deficit in nictation behavior and that optogenetic activation of ASG did not increase the nictation ratio (Supplementary Fig. 5a, b), suggesting that ASG neurons are necessary, but not sufficient, for nictation behavior and that ASG neurons may not be a member of the direct circuit of nictation. In summary, our investigation of individual neuronal changes within the connectome revealed that the changes in morphology of neurons in dauer are closely related to connectivity changes, which could explain the stage-specific behavioral changes.

## The networks of dauer and adult share common structural properties

Next, we compared the entire dauer connectome with connectomes of other stages systematically using graph theoretical analyses, seeking global characteristics of the dauer connectome. We aforementioned the dauer connectome has a similar number of connections with adult

connectome (Fig. 2a). Consistently, the average in- and out- degrees of neurons both exhibits increase across developmental stages and the dauer connectome shows similar values with adult connectomes (Fig. 5a). Furthermore, when we look at the degree distributions of each stage, the distributions decay roughly exponentially as revealed by linear decrease in the semi-log plot (Fig. 5b). The exponential decay implies that the network mainly consists of peripheral neurons and only a few small hub neurons such as AIB, RIA, and RIH. In addition, the distributions are shifted toward larger values and tails tend to become longer for dauer and adult-stage networks (Fig. 5b), suggesting hub neurons tend to have more connections in the dauer and adult stages. As dauer and adult stages consist of more connections, it leads to shorter mean path length (Fig. 5c)[20] and higher clustering coefficient (CC)[21], which measures the probability of the pair of neurons that are connected to a common neuron being connected to each other (Fig. 5d). The other basic network features are shown in

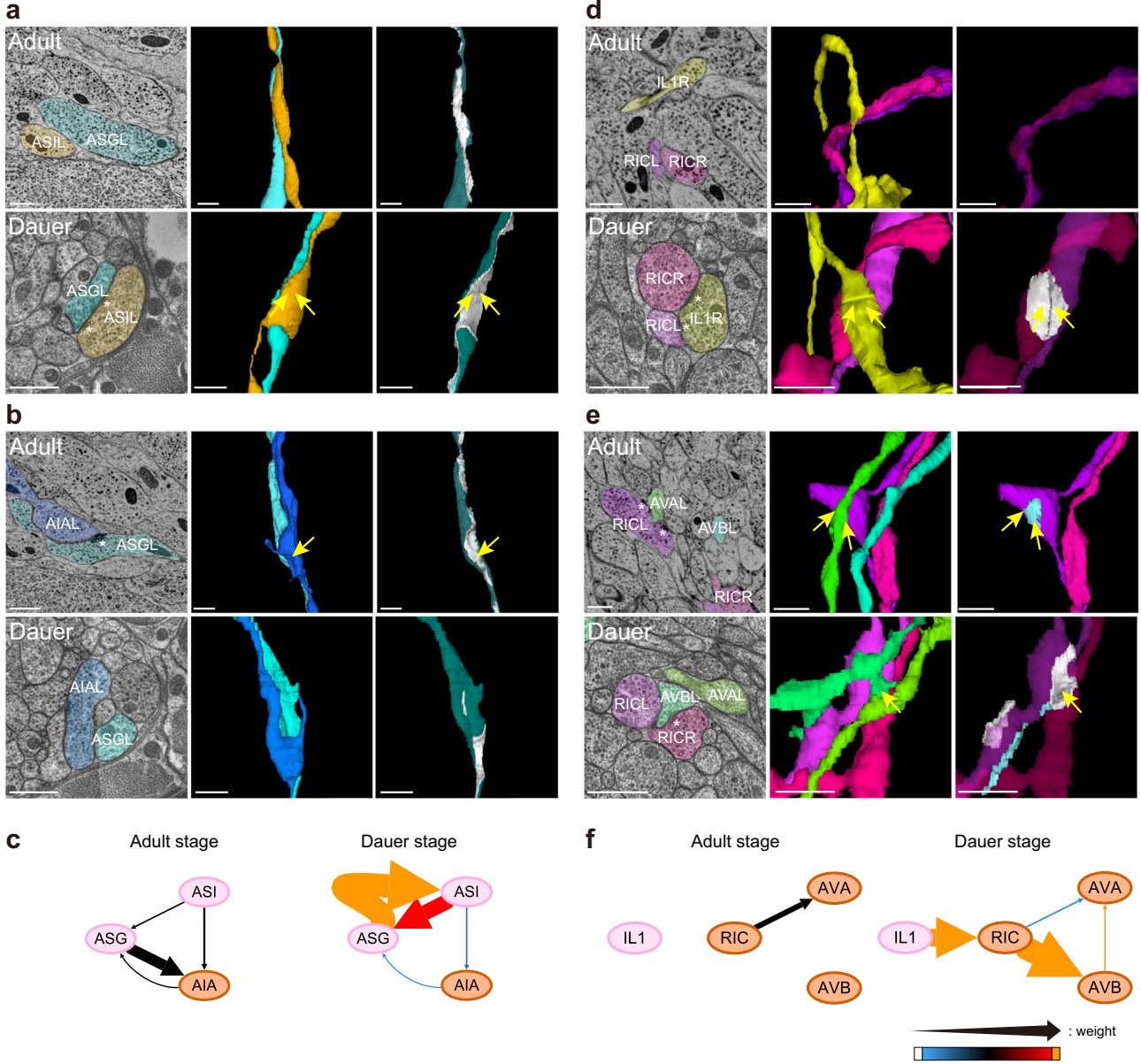

**Fig. 3 | Morphological changes and connectional changes are mutually associated. a** EM cross-section view (left) and 3D volumetric reconstructions of presynaptic and postsynaptic neurons (middle) at the location of ASG-ASI contact where synapse exists (right, contact area marked as white) in adult-2 (top row) and dauer (bottom row). Contact area between ASG and ASI was significantly larger in dauer resulting in dauer-specific synapses (yellow arrows). Note that EM cross-section views of two stages may look different as two datasets have been sectioned in different orientations. **b** Same with **a** for ASG-AIA contact. ASG-AIA contact area is significantly reduced in dauer while ASG-AIA contact area is distinctive resulting in larger synapse (yellow arrow). **c** Wiring diagram of neurons shown in **a** and **b** in adult-2 stage (left) and dauer stage (right). Width and color of the arrow follows the same rule from Fig. 2d. Larger contact between ASG and ASI results in strong reciprocal connections and smaller contact between ASG and AIA results in no

connection in dauer. **d** Same with **a** for RIC-IL1 contact. IL1-RIC contact exists in the dauer stage while they do not make contact in the adult-2 stage resulting in dauer-specific synapses (yellow arrows). **e** Same with **a** for RIC-AVA (light blue) and RIC-AVB (white) contacts. Area of RIC-AVA contact decreased in the dauer stage, leading to lack of synapses in the dauer stage while RIC-AVA synapses (yellow arrows) exist in the adult-2 stage. Area of RIC-AVB contact increased in the dauer stage resulting in dauer-specific synapses (yellow arrow). **f** Wiring diagram of neurons shown in **d** and **e** in adult-2 stage (left) and dauer stage (right). Contact between RIC-IL1 and RIC-AVB results in new connections and lack of contact between RIC-AVA results in no connection in dauer. **a**–**e** Scale bars: 500 nm (EM images) and 1 μm (3D views), asterisk: active zone, yellow arrow: stage-specific synapse.

Supplementary Fig. 6[22–26]. Overall, the dauer connectome shares common global network structures with adult connectomes.

### Dauer-specific connections contribute to increased clustering within motor neurons

While the dauer and adult stages share common features in primary network properties, we further explored whether there are differences

between dauer and adult stages in subnetwork (a subset of the whole network with pre- and postsynaptic neurons as one of sensory, inter-, or motor neurons) properties. We first divided the connections into 9 types according to the types of pre- and postsynaptic neurons (i.e., sensory, inter-, motor neurons) and measured the out-degree of neurons in each connection type (Fig. 6a, Supplementary Fig. 7a[4]). The out-degree of each connection type did not differ in most connection

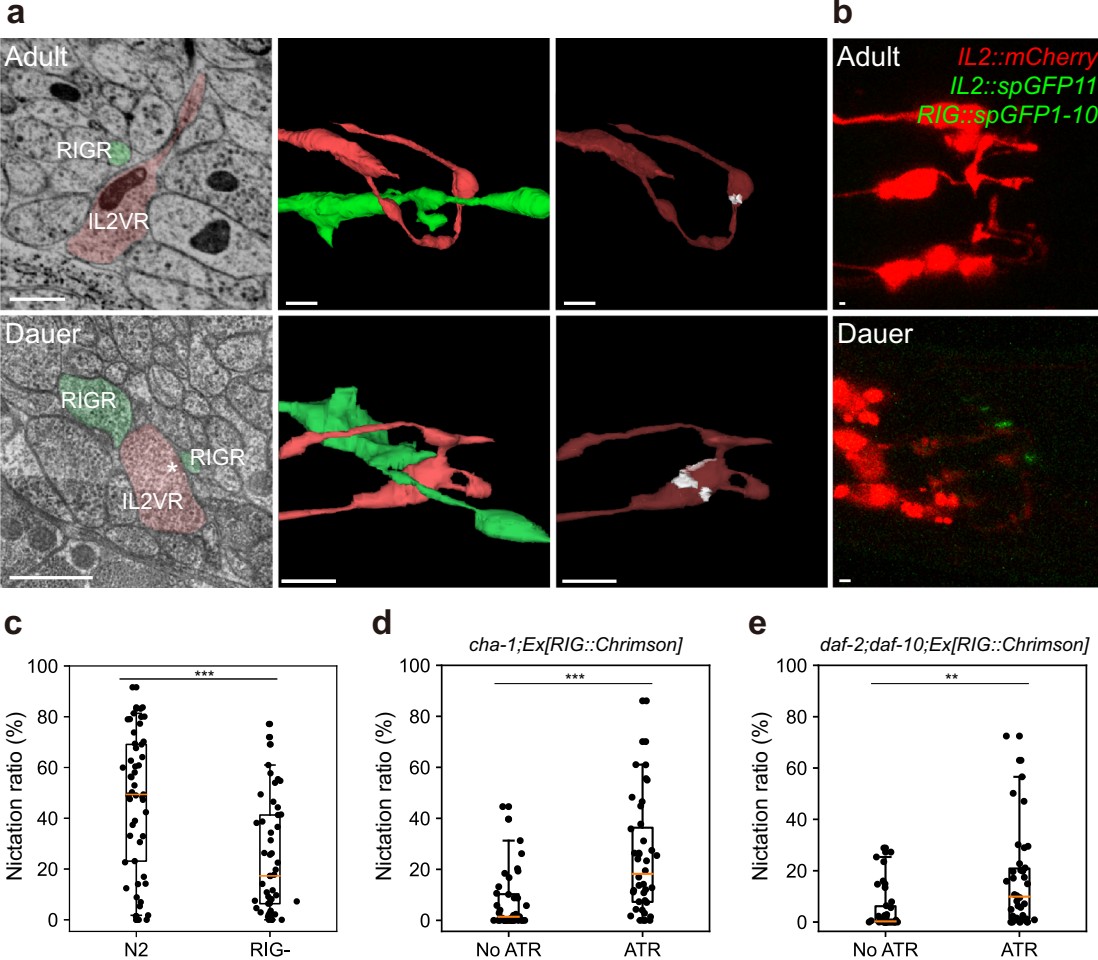

**Fig. 4 | Connectivity changes in dauer neurons are correlated with unique behavior in dauer. a** EM cross-section view (left) and 3D volumetric reconstructions of presynaptic and postsynaptic neurons (middle) at the location of IL2-RIG contact (right, contact area marked as white) in adult-2 (top row) and dauer (bottom row). **b** Fluorescence images of IL2 neurons and GRASP signal in adult (top) and dauer (bottom) stages. IL2 neurons were marked with mCherry using F28A12.3 promoter. Split GFP11 was tagged with IL2 neurons using F28A12.3 promoter, and split GFP1-10 were tagged with RIG neurons using *twk-3* promoter. **c** RIG ablated lines show significantly lower nictation ratio than N2 (individual nictation test;

$n = 49$; two-sided unpaired t-test; ***$p = 1.8×10^{-5}$). **d** RIG-activated *cha-1* mutants exhibit significantly higher nictation ratio than *cha-1* mutants (individual nictation test; $n = 40$; two-sided unpaired t-test; ***$p = 4.9×10^{-5}$). **e** RIG-activated *daf-2;daf-10* mutants exhibit significantly higher nictation ratio than *daf-2;daf-10* mutants (individual nictation test; $n = 38$; two-sided unpaired t-test; **$p = 0.0016$). **a** Scale bars: 500 nm (EM images) and 1 μm (3D views), asterisk: active zone. **b** Scale bars: 1 μm. **c-e** Orange line: median, box: interquartile range, whiskers: 5th and 95th percentiles.

types except for inter → motor connections, where dauer shows significantly less out-degree and motor → motor connections, where dauer shows significantly greater out-degree. This indicates notable proportion of dauer-specific connections (Fig. 2c) contribute to addition of connections within motor neurons. Thus, we inspected how these additional connections affect the network characteristic. The same measure for in-degree of neurons can be found in Supplementary Fig. 7b.

To characterize the motor → motor connections which showed marked increase, we considered three types of subnetworks, each consisting of connections within a single type of neuron. The motor subnetwork was analyzed in comparison to sensory and interneuron subnetworks. We first performed three-node motif analysis in each subnetwork and discovered that the proportions of the triangular motifs, where all three neurons in a motif are connected to one another, were higher compared to the rest of the motifs (Fig. 6b, Methods). As a reference, the proportions of triangular-motifs were also measured in the equivalent Erdős–Rényi (ER) random networks with the same number of nodes and average degrees. The motor subnetwork had highest proportion in dauer, indicating the

connections of motor neurons of dauer are wired to make more triangular motifs compared to motor neurons of adult (Fig. 6b). Motor neurons had the highest proportion compared to other types of neurons in dauer while this trend was not found in adults (Fig. 6b). This is not due to the higher connection probability, since the equivalent ER random network could not reproduce the similar difference in triangular-motif proportion between dauer and adults in motor subnetwork (Fig. 6b). This evidence highlights that dauer-specific connections between dauer motor neurons are programmed to make an exceptionally large number of triangular motifs.

As a validation, we computed the CCs of neurons in the three subnetworks and the neurons in the dauer motor subnetwork exhibited significantly larger CC than adult motor subnetworks while there were no differences in sensory and interneuron subnetworks (Fig. 6c). This further supports the conclusion that dauer motor neurons are specifically connected for local clustering. The subnetwork diagrams of major motor neurons for adult and dauer exemplify that the dauer-specific connections enhance the proportion of triangular motifs and the clustering among motor neurons in dauer (Fig. 6d).

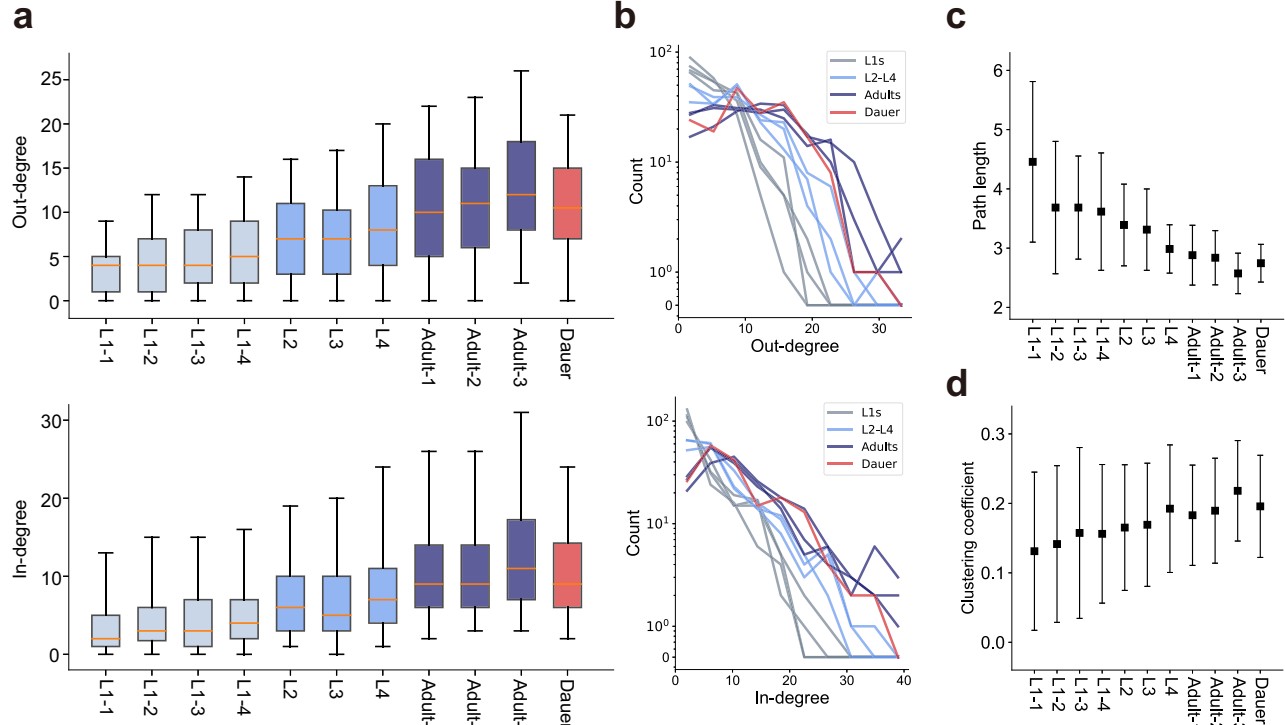

**Fig. 5 | The networks of dauer and adult stages share common structural properties. a** Out- (top) and in-degrees (bottom) of neurons (*n* = 180; orange line: median, box: interquartile range, whiskers: 5th and 95th percentiles) across development. Boxes are colored by developmental stages (gray: L1s, light blue: L2 to L4, dark blue: adults, red: dauer). **b** Out- (top) and in-degree (bottom) distributions of different developmental stages. Line color follows the same rule in **a**. **c** Mean path lengths of neurons (*n* = 140, 142, 143, 145, 153, 152, 168, 159, 160, 177, 164; mean ± SD) across development. **d** Clustering coefficients of neurons (*n* = 155, 161, 158, 164, 172, 174, 178, 180, 180, 179, 179; mean ± SD) across development.

## Dauer-specific rewiring of sensory neuron connectivity

Unlike motor neurons, other types of neurons did not show significant increase in out-degree (Fig. 6a). Considering the high proportion of dauer-specific connections (Fig. 2c), we hypothesized the connection target distribution could differ for sensory neurons and interneurons leading to rewiring in connectivity.

We first analyzed the number of common connections between the datasets. We found that the number of common connections in sensory neuron output connections between dauer and adult datasets is significantly smaller than the number of common connections between adult datasets (Fig. 6e, Supplementary Fig. 7c, d, Supplementary Fig. 8a). On the other hand, the number of common connections in inter- and motor neuron output connections between dauer and adult datasets did not exhibit differences with the number of common connections between adult datasets (Fig. 6e, Supplementary Fig. 7b, c, Supplementary Fig. 8a, b). This means the smaller number of common connections from sensory neurons is a stage-specific phenomenon in contrast to inter- and motor neurons.

To strengthen our argument for rewiring of sensory neurons in terms of target variation, we examined "connection similarity (CS)", which measures the similarity of connection targets of neurons between two stages (Methods). We found that CS between dauer and adult stages are significantly lower than CS among adult stages in sensory neurons while this is not the case for interneurons (Fig. 6f, Supplementary Fig. 8c, d). This result implies that output connections of sensory neurons are dauer-specifically rewired compared to adults while both have similar number of connections. This result is consistent with results regarding the number of common connections (Fig. 6a, f, Supplementary Fig. 7c, d, Supplementary Fig. 8c). For interneurons, CSs exhibited a similar level for every pair of datasets indicating it is due to variability between datasets rather than dauer-specific phenomenon (Fig. 6a, f, Supplementary Fig. 8c). For motor neurons, CSs of dauer and adults were lower than CSs between adults unlike the results regarding the number of common connections. This provides additional evidence that dauer-specific connections contribute to additional connections on top of common connections for motor neurons in dauer (Fig. 6a, f).

In conclusion for the graph theoretical analyses, there exists increased clustering among motor neurons and significant rewiring in the connectivity of sensory neurons in the dauer connectome, creating differences from adult connectomes (Fig. 6d, g, h).

## Discussion

Complete stage-wise comparison of neural circuits at the single synapse level is becoming feasible as EM connectome datasets in all developmental stages have been published[2,4,5,15,16] except for the alternative stage, dauer. To solve the last piece of the puzzle, we have reconstructed the first dauer nerve ring connectome from EM images, consisting of dense volumetric reconstructions of both neuronal and muscle cells, and their connectivity acquired using deep learning. Using the publicly available resource, we have found that various morphological changes of neurons occur at the dauer stage and these morphological changes are related to connectivity changes. Also, we have shown that dauer-specific connections are associated with the unique survival behavior, nictation. Lastly, we found that the neural network of dauer exhibits rewired sensory connections and increased clustering among motor neurons.

As we have reconstructed only one dauer connectome, there exists a possibility that the features described in this study could be due to individual variability. We have conducted fluorescence imaging to validate certain features (e.g., IL2 morphological changes) also exist in other dauer animals but we could not validate all features. Gap junctions, another important component in the neural circuit, are not included in this study as improvements are required to effectively

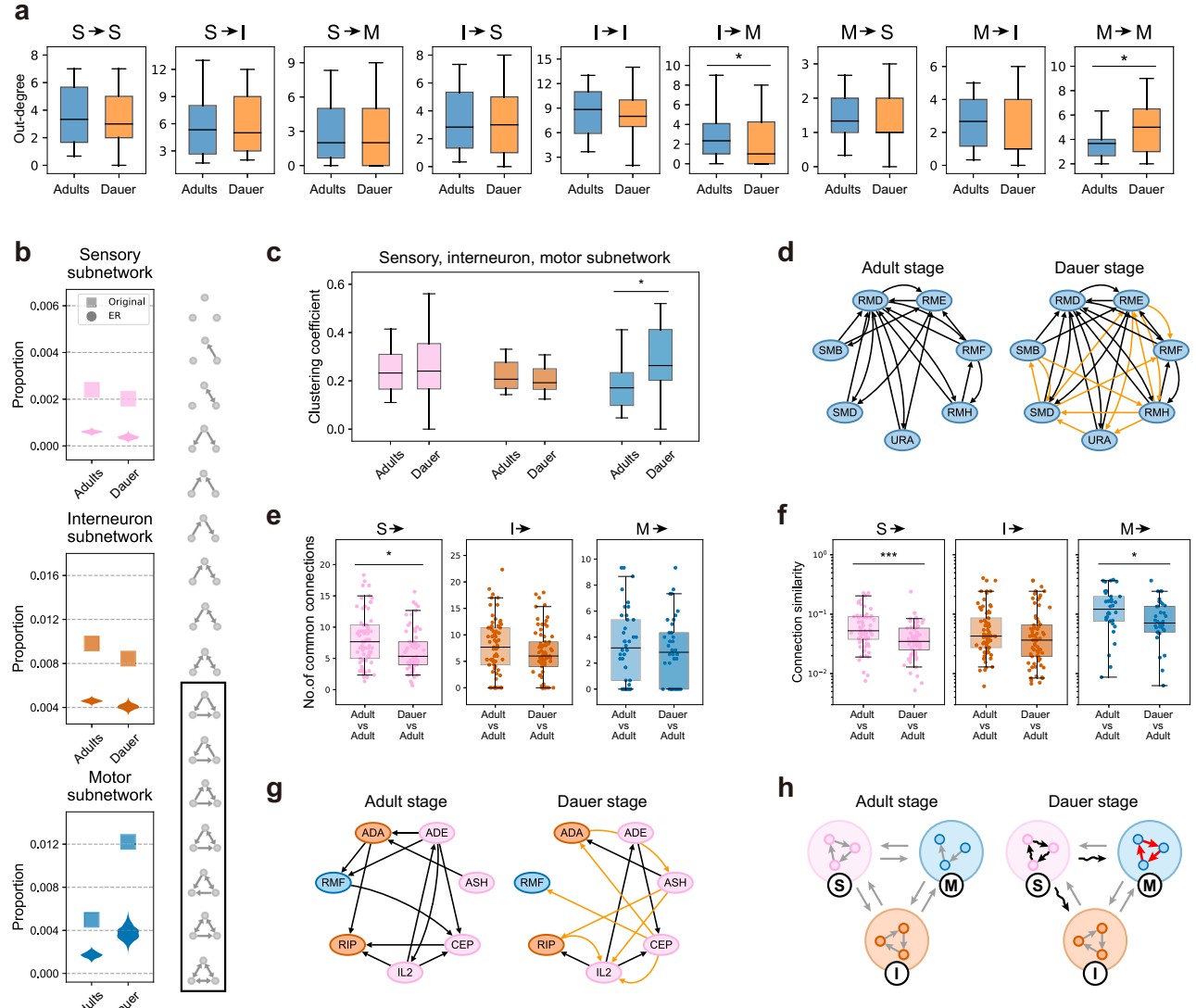

**Fig. 6 | Dauer connectome exhibits rewired sensory connectivity and increased clustering within motor neurons. a** Out-degrees of neurons for different connection types ($n = 60$ (I→M), $n = 27$ (M→M), *$p = 0.042$ (I→M), 0.042 (M→M)) in adults (blue) and dauer (orange). Neurons without any connection in any of the datasets were excluded. S: sensory neuron, I: interneuron, M: motor neuron. **b** Observed proportion of triangular motifs (motifs in black rectangle; square) in sensory (pink), interneuron (orange), motor (blue) subnetworks in adults and dauer and their expected distributions in corresponding Erdős−Rényi random networks (violin plot). **c** Clustering coefficients of neurons in sensory (pink), interneuron (orange), motor (blue) subnetworks in adults and dauer (*$p = 0.017$). Neurons with less than 2 connections in the subnetwork were excluded. **d** Wiring diagrams of representative motor neurons in adult-2 and dauer stages. Dauer-specific connections (orange) add more connections in the motor subnetwork, leading to higher local clustering. **e** Number of common output connections from sensory (pink),

inter- (orange), motor (blue) neurons between adults (left; $n = 69$ (S), 73 (I), 38 (M)) and that between dauer and adults (right; $n = 69$ (S), 73 (I), 38 (M); *$p = 0.022$). **f** Connection similarity of output connections from sensory (pink), inter- (orange), motor (blue) neurons between adults (left; $n = 69$ (S), 73 (I), 38 (M)) and that between dauer and adults (right; $n = 69$ (S), 73 (I), 38 (M); *$p = 0.010$, ***$p = 2.2 \times 10^{-4}$). **g** Wiring diagrams showing example rewired sensory neurons in adult-2 and dauer stages. Dauer-specific connections (orange) add new connections while losing connections that exist in the adult, leading to rewired sensory connectivity. **h** Summary diagram illustrating differences between dauer and adult networks. Black curved arrows represent the remodeling of sensory outputs. Red arrows indicate increases in clustering. **a, c, e, f** Black line: median, box: interquartile range, whiskers: 5th and 95th percentiles, statistics: two-sided Wilcoxon rank-sum test. **a, b, c** Values in adult are computed by averaging over three adult datasets. **e, f** Values are computed by averaging over all pairs of datasets.

identify them with the provided data. With complete gap junction information, it would be not impossible to predict a more precise circuit of behaviors. So far, we have attempted to identify gap junctions that are obvious and the data is available upon request. Despite the above caveats, it is still the only publicly available dauer connectome dataset so we hope this resource could help the community in stage-wise comparative study of *C. elegans*.

Morphological changes associated with synaptic changes must not be a mere coincidence but must be an event with concrete physiological advantages, because morphological remodeling is a highly

energy-consuming process[27,28]. The local morphological changes are permissive for connectional changes, and these changes can lead to valuable behavioral changes, even without changes in the number of neurons involved. The changes in morphologies and connections can be identified by comparative connectomics.

The original question that Brenner posed, how the nervous system governs the behavior, can now be recast as the relationship between dauer-specific connections and unique behaviors of the dauer, which can be further explored by future experimental studies. One example is $CO_2$ sensing in different developmental stages. Our

data sheds light on the observation that in the dauer, unlike in the adult, AIY is not required for $CO_2$ response[29]. Our data shows that the BAG → AIY connection is absent in the dauer[11]. Now, it is possible to focus on additional stage-specific connections and check how the neuronal functions differ in terms of $CO_2$ sensing using optogenetics and calcium imaging like the IL2-RIG experiments we have conducted. Another example is the connection changed from RIC → AVA in other larval stages to RIC → AVB in the dauer. Proper analyses and experiments will reveal the significance of this shift in dauer-specific motility. The absence of pharyngeal pumping in dauer stage is also worth studying. The RIP neurons are the only neurons that connect the somatic and pharyngeal connectome and all of their connections are dramatically changed in dauer (Supplementary Data 1). It would be interesting to examine possible roles of RIP neurons in the regulation of pumping of dauer. It is not trivial to discuss the motionless trait of dauer but we could get some hints from the network properties of motor neurons. Motor neurons are somewhat isolated from sensory and interneurons in dauer (Fig. 6a, Supplementary Fig. 6b). A testable hypothesis is that the overall isolation of motor neurons in terms of network connectivity can be used as a strategy to conserve energy by rewiring the motor output to remain unresponsive, except in the presence of crucial input signals. As these examples demonstrate, we believe that our new dauer connectome could lead to new discoveries of neural mechanisms for dauer-specific functions.

Along with other developmental stages, dauer neural network also exhibits its own form of variability (Fig. 2c, d)[2,4,5,15,16]. As the first distinctive characteristic of dauer neural network, the enhanced clustering and augmented triangular motifs (Fig. 6b–d), can allow faster signal transduction by making shortcuts, can enhance the robustness in the signal processing by providing an alternative path, or can amplify the signal by making a cyclic, recurrent connection[30]. As the nictation seems to be initiated in the nose, our nerve ring connectome could be potentially useful in elucidating the mechanism of nictation.

As a second distinctive characteristic, output connectivity of sensory neurons is dauer-specifically rewired compared to adults (Fig. 6e–g). While sensory connectivity is developmentally dynamic, strengthening or weakening existing connections, in the normal reproductive stages[2,4,5,15,16], dauer sensory neurons send signals to different target neurons (Fig. 6e–g), which could explain the differences in chemosensation. While dauers still show normal chemosensory responses to many chemical attractants as other reproductive stages, dauers also exhibit specificity[6,31]. For example, it has been reported *C. elegans* in the dauer stage respond differently to $Na^+$ and $CO_2$[7–12]. These qualitatively different network structures may support the survival strategies of the dauer.

Unlike sensory and motor neurons, dauer interneurons show similar variability in connectivity to variability within different adult datasets (Fig. 6e–g). Wiring of interneurons are known to be developmentally stable, serving as the core central processor[2,4,5,15,16]. Our results imply stability of interneuron connectivity could be conserved in the dauer stage as well, sharing the same central processor. Neurons classified as interneurons by our definition[4] included modulatory neurons defined by Witvliet et al.[5], which have many variable connections. For fair comparison, we have rerun the analysis without the modulatory neurons and discovered CS among datasets did not show significant difference (Supplementary Fig. 9), maintaining the hypothesis that the stability of interneuron connectivity is conserved in dauer.

In summary, as the last piece of the nematode connectome, we have reconstructed the first dauer nerve ring connectome from EM images and have shown that the dauer connectome is quantitatively similar to and qualitatively distinct from the adult connectome. Our dauer connectome resource will open up many possibilities to study how the neurons function differently due to differences in the neural circuit, and vice versa.

## Methods

### Experimental model and subject details

*C. elegans* were cultured and handled at 20 °C according to standard methods[32]. *C. elegans* Bristol strain N2, *cha-1*(n2411), *daf-2*(e1370), *daf-10*(e1387) lines were used, obtained from the Caenorhabditis Genetics Center (CGC)[12]. JN2114 (Is[*gcy-21p*::mCaspase, *gcy-21p*::mCherry, *lin44p*::GFP]) was provided by Iino Yuichi's lab[33]. EAH813 (Ex[*twk-3p*::casp-3(p17); *twk-3p*::casp-3(p12); *myo-2p*::mCherry]) was provided by Elissa A. Hallem's lab[34]. For TEM fixation, *C. elegans* Bristol strain N2 were used.

### Dauer induction

For TEM fixation, seven to eight young adult larvae were transferred to synthetic pheromone plates containing agar (10 g/L), agarose (7 g/L), NaCl (2 g/L), KH2PO4 (3 g/L), K2HPO4 (0.5 g/L), cholesterol (8 mg/L), and synthetic pheromone-ascaroside 1, 2, 3 (2 mg/L each)[35,36] seeded with *Escherichia coli* OP50 at 25 °C for dauer induction[37]. After 4 days, dauers were induced and morphologically identified by their radially constricted bodies and sizes. For microscopy and nictation assay, only synthetic pheromone-ascaroside 2 (3 mg/L each) were used for dauer induction.

### Fixation and staining for dauer larvae

Given the thicker cuticle on the dauer, we utilized an enhanced "OTO" protocol[38] to conduct multiple rounds of osmium tetroxide exposure following a high-pressure freeze fixation to add extra layers of osmium stain prior to finishing a more regular counterstain with uranyl acetate and lead aspartate. The cryofixation by high-pressure freeze and freeze-substitution (HPF-FS) better preserves membrane quality, while the OTO adds to the final contrast compared to a single exposure to osmium.

Live dauers were immediately transferred into a bacterial slurry in a type A hat, matched to a sapphire disk to close the volume, backed by a type B hat (flat side towards sapphire disk) and mounted into the specimen holder of a Bal-tec HPM010 High Pressure Freezer. The samples were frozen under high pressure and moved to the first fixative, 5% glutaraldehyde, 0.1% tannic acid in acetone, at liquid nitrogen temperature. Multiple samples from the same run could be fixed in the same tube to conserve space in the freeze substitution device, and assure that portions of samples are well-frozen.

Frozen samples were moved into a Leica freeze substitution device at a holding temperature of −90°C and held at that temperature for 64 hours, then warmed to −60°C in 5°C steps and held for another 48 hours. During this time, samples were washed several times in dry cold 98% acetone, then shifted to the second fixative, 2% osmium tetroxide, 0.1% uranyl acetate, 2% double-distilled (DD) water in acetone. After 48 hours, temperature was then raised slowly to −30 °C in 5°C steps and held at −30 °C for 16 hours. Then, it was immediately followed by four washes in 98% cold acetone for 15 minutes each. Samples were removed from the freeze substitution unit and put onto ice, then transferred to 0.2% thiocarbohydrazide in acetone for one hour at room temperature. After four 15-minute washes in 98% acetone, the samples were moved to the third fixative, 2% osmium tetroxide, 0.1% uranyl acetate, 2% DD water in acetone on ice for one hour. After three 10-minute washes in 98% acetone at room temperature, the samples were stained in 1.5 ml of filtered 2% lead aspartate in acetone for 15 minutes at room temperature. The samples were transferred to microporous specimen capsule (EMS, cat # 70187-21) for further handling of multiple worms without sample losses.

The samples were washed four times in 100% acetone for 15 minutes each at room temperature. Stepwise infiltration with resin:acetone dilutions (1p:2p and 2p:1p) was performed. Individual animals were mounted on Lab-Tek chamber slides with cover glass slide (EMS, cat # 70411) as molds. Then, cover glass was removed and the samples were cured in a 60 °C oven for 48 hours, which makes it

possible to cut out single animals from a thin layer of resin and Aclar to remount for thin sectioning.

## Electron microscopy image acquisition for dauer larvae
RMC PowerTome XL Ultramicrotome (Boeckeler Instruments, Inc.) and a 45-degree Ultra Diamond Knife (Electron Microscopy Sciences, cat # 40-US) were used for sectioning at a speed of 1 mm/s to cut 50 nm serial sections. The sections were collected onto Formvar-coated slot grids by hand[39]. The sections were post-stained, first with 2% uranyl acetate for 20 minutes, and then a 1:5 dilation of Reynold's lead citrate for 5 minutes.

Transmission electron microscopy imaging was conducted on a JEOL JEM-1400Plus microscope, equipped with a Gatan Orius SC1000B bottom mount digital camera. Using Serial EM software, we collected about 70 individual images per thin section.

## Stitching and alignment
Stitching and alignment for dauer EM images were done using TrakEM2 software[40]. First, image patches were normalized to have similar contrast and brightness. Scale-invariant feature transform (SIFT[41]) was applied to identify local features in the overlapping area of the adjacent image patches. Based on identified features, optimal affine transformation was computed, which minimizes least squares error, then applied to stitch image patches.

To align the stitched image sections, the total image stack was divided into multiple chunks of image sections and each chunk was aligned separately to prevent large distortion. For each chunk, image features were manually selected from the first and last sections, then the affine transformation parameters that minimize least square error were computed. Transformations for intermediate sections were computed by interpolation according to the section's distance from the first and last sections. Affine transformation with manual features was applied to align the chunks. Lastly, elastic alignment in TrakEM2 was applied for the final alignment of the image stack.

## Volumetric reconstruction and cell identification
All neurons, muscle cells, and glia in the EM volume were volumetrically segmented. Using VAST[42], annotators manually traced each neuron and colored the cross-section of the neuron. The direction of the left-right axis was determined with the asymmetric ventral nerve cord. The neurons were identified by anatomical features such as location of soma, neurite trajectory, and distinctive morphological traits using other published EM datasets as references[2,5,15,16]. Neuronal types were defined by classification used in Cook et al.[4].

## Chemical synapse detection and synaptic partner assignment
Two areas of the volume were selected from the EM volume and 10 sections from each area were chosen to be annotated for the ground truth labels. Human annotators painted presynaptic active zones, which were darkly stained pixels near the membrane, using VAST[42] and the results were reviewed by another annotator. Ground truth labels covering an area of 9053 μm² were generated. The labels were divided into train (7340 μm²), validation (874 μm²), and test (839 μm²) datasets.

The network used for the chemical synapse detection was adopted from 2D symmetric U-Net architecture[43]. The architecture was composed of five layers with a number of feature maps, 16, 32, 64, 128, 256 from the top most layer to the bottom most layer, respectively. For each step in downsampling and upsampling layers, the architecture consisted of three 3 ×3 non-strided same convolutions. In the downsampling layers, max pooling was used to downsample by a factor of 2 in each layer. In the upsampling layers, a transposed convolution layer with nearest neighbor interpolation followed by 2 ×2 non-strided convolution was used to upsample by a factor of 2 in each layer. Skip connection was included in every layer which concatenates feature maps at the same level of the left-hand side to the output of the

transposed convolution layer. Instance normalization[44] and rectified linear unit (ReLU) was added after each 3 ×3 convolution. At the end of the symmetric network, 3 ×3 non-strided same convolution and sigmoid function was applied to produce the same-sized output image, where each pixel value represents probability of each pixel belonging to the presynaptic active zone. The network takes an input EM image that has patch size of 576 ×576 pixels in 2 ×2 nm² resolution and outputs an image with size of 512 ×512 pixels in the same resolution. Note that the input image is 64 pixels larger in each dimension as 32 pixels from the edges have been cut off to avoid inaccurate predictions near the borders.

The network has been trained using an Adam optimizer[45] and using binary cross entropy as a loss function. The learning rate has been kept constant at 0.0005. Horizontal and vertical flip and multiple of 90-degree rotation were applied on training samples as data augmentation. Also, brightness and contrast augmentation have been applied.

From the predicted probability map of active zones, we thresholded the image with a pixel threshold of 90 / 255 (35%) and generated binarized prediction images of active zones. To reconstruct active zones in 3D, we applied connected components with connectivity of 26. The errors in resulting reconstructions were corrected manually.

For each active zone, we ran Monte Carlo simulation of neurotransmitter diffusion[5] to identify potential synaptic partners. We assigned synapses sizes for each synaptic partner according to the proportion of particles that reached its segment.

## Evaluation of synapse detection model
Two test volumes of size 1024 ×1024 x 1600 nm³ have been randomly selected from regions that were not covered in the training data. To generate the ground truth for the test volumes, two annotators have independently labeled the active zones in the volumes. Then, both annotators and the third annotator reviewed the volume and proofread the annotations.

We compared the predicted active zone segments and counted predicted active zone segments that overlap with ground truth segments (true positive), predicted segments that do not overlap with any ground truth segment (false positive), and ground truth segments that do not overlap with any predicted segment (false negative). Since we are not able to define active zone segments in the undetected region, true negatives could not be quantified.

## Generation and modification of transgene constructs
All constructs in this study were created using Gibson assembly cloning kit (E5510; New England Biolabs). DNA fragments were inserted into the GFP vector (pPD95.77) and mCherry vector (pPD117.01; modified to mCherry red fluorescence). For transcriptional fusion constructs, we inserted the promoter region of relevant genes into vectors. The indicated numbers are relative to the start codon ATG for each gene, respectively: *egas-4p::GFP*; *klp-6p::GFP*; *twk-3p::Chrimson::*SL2::*mCherry, lron-4p::Chrimson::*SL2::*mCherry*[46]. For GRASP, we used *F28A12.3p*::NLG-1::spGFP11 and *twk-3p*::NLG-1::spGFP1-10. Promoters were substituted from *flp-18p*::NLG-1::spGFP11 and *unc-4p*::NLG-1::spGFP1-10, gifts from Cori Bargmann & Kang Shen (Addgene plasmid # 65827 and #65828)[19]. All details of transgene constructs are available upon request.

## Generation of transgenic animals
To generate transgenic animals, we microinjected DNA and plasmid constructs into the gonad of young adult hermaphrodite as a canonical method[47]. We used following co-injection marker with indicated concentration for transgenesis to isolate transgenic progeny of microinjected hermaphrodite: *unc-122p::ds-RED* (25 ng/ul; red fluorescent coelomocyte), *unc-122p::GFP* (40 ng/ul; red fluorescent coelomocyte) and *myo-3p::mCherry* (50 ng/ul; red fluorescent muscle). Plasmid DNAs

used for microinjection were extracted and purified using a plasmid purification kit. All details of transgenic animals are available upon request.

## Fluorescence microscopy

Images of worms were acquired using the confocal microscope (ZEISS LSM700; Carl Zeiss). For imaging, transgenic worms were paralyzed with 3 mM levamisole and mounted on 3% agar pads. Each strain was imaged at least 20 times, and one representative image was included in the manuscript. To examine the morphological and synaptic changes of neurons, we visually inspected neurons by maximum intensity projection using ZEN software (Carl Zeiss) and image averaging using Matlab.

## Contact matrices and contact area

In non-dauer datasets except for adult-1 dataset, which does not have segmentation images, segments in the region from where the nerve ring begins to the entry point of the ventral nerve cord (approximately 12 - 13 μm length in L2 and L3) were used to compute contact matrices for fair comparison between different size image volumes. Reasonable background masks were obtained by manually annotating the dilated results of all segments. Each segment was dilated one at a time in pseudo-random order repeatedly until the background masks were fully covered. As non-dauer datasets do not have cell bodies and long amphid processes segmented, these segments were removed from the dauer segmentation images when computing contact matrix.

To measure the contact area between cell segments, contact direction (i.e., x-y, x-z, or y-z) was identified for each voxel in the contact region. For each contact voxel, the area of the square orthogonal to the contact direction was measured, and contact area was calculated as the sum of the areas for all contact voxels, according to the resolution. For fair comparison of contact areas between datasets, contact areas have been normalized by the standard deviation of connection weights without the top 5th percentile to remove the bias due to the big outliers. Then, these contact areas and normalized contact areas have been assigned to corresponding locations in the 180 ×180 (neurons) common contact matrix structure for all datasets to create contact matrices.

## Connectivity matrices and connection weights

Connection weights between a connected pair of neurons were computed as the sum of all synapse sizes of synapses between pre- and postsynaptic neurons of the pair. For fair comparison of connection weights between datasets, connection weights have been normalized by the standard deviation of connection weights without the top 5th percentile to remove the bias due to the big outliers. Then, these connection weights and normalized connection weights have been assigned to corresponding locations in the 180 ×180 (neurons) common connectivity matrix structure for all datasets to create connectivity matrices.

## Wiring diagrams

Network diagrams were drawn using Cytoscape[48]. In the diagram, edges with normalized connection weights below the average of individual synapse weights ($w = 0.2728$) were excluded. Also, connections between neurons and CEPsh cells and autapses, synapses to itself, have been excluded from the diagram. Connections are drawn as arrows, indicating the direction of connections, and the width of arrow represents the connection weight. In Fig. 2d and Supplementary Data 1, the color of the edges indicates the changes in normalized weights in the dauer stage. Red and blue indicate increase and decrease in the dauer stage respectively. Orange indicates dauer-specific connection.

## Stage-conserved and stage-specific connections

Connections included in the intersection of all 11 datasets (Fig. 2b) were assigned as stage-conserved connections. In each dataset, the connections that only exist in the dataset and cannot be found in any of the other 10 datasets were assigned as stage-specific connections.

## Nictation assay

First, we created a micro-dirt chip by pouring a 4% agar solution onto a PDMS mold[37]. The solidified chip was then detached from the PDMS mold and dried for 90 min at 37 °C. For nictation assays, individual dauer were observed for 1 min and measured nictating time. The nictation ratio was calculated from the ratio of nictating time and total observation time. Quiescent dauers were excluded from measurements. Assays were carried out at 25 °C with a humidity of 30%[12,37]. For optogenetic experiments, young adult worms expressing Chrimson transgenes were transferred to either normal OP50 pheromone plates or OP50-retinal pheromone plates containing 1 mM all-trans-retinal (Sigma). The transgenic dauers derived from the pheromone plate were transferred to a micro-dirt chip and their behavior was recorded for approximately 1 minute under green light (510-560 nm) on a Leica M205 FA microscope. The behavior of a dauer from the OP50-retinal pheromone plate and a dauer from the normal OP50 pheromone plate was recorded alternately, and the nictation ratio was quantified from the recorded movies.

## Network properties

Total of 180 neurons, excluding CAN neurons as they have no connections, were used to compute network properties. Among 180 neurons, 69 were sensory neurons, 73 were interneurons, and 38 were motor neurons. The analysis was restricted to a network of neuronal cells, excluding muscle cells and end organs, as non-neuronal cells have inconsistencies between datasets.

Out- and in-degrees of a neuron was calculated by counting the number of postsynaptic partners and presynaptic partners, respectively. Out-degree for each connection type was calculated by counting the number of postsynaptic partners in that subnetwork. Neurons without any connection in any of the dataset were neglected when computing out-degrees of different connection types.

Path length between a pair of neurons is computed by counting the minimum number of steps from the source neuron to the destination neuron.

Clustering coefficient (CC) measures the probability of the pair of neurons that are connected to a common neuron being connected to each other. CC was defined as

$$CC(u) = \frac{2T(u)}{deg(u)(deg(u) - 1)} \tag{1}$$

where $deg(u)$ is the number of edges connected to the neuron $u$ regardless of their directions. $T(u)$ indicates the number of triangles via node $u$, which occurs when the two neurons connected with neuron $u$ are connected to each other. Nodes with less than two edges were excluded from the analysis as they have no chance of forming a triangle. Besides measuring the CC of the whole network, CC has been also measured in the sensory, interneuron, and motor subnetworks which only consist of connections within the same type of neurons ($n = 63$ (S, adult), 64 (S, dauer), 72 (I, adult), 73 (I, dauer), 31 (M, adult), 31 (M, dauer)).

## Number of common connections

Number of common connections measures the number of outgoing connections of a neuron that exist in both datasets being compared. The number of common connections has been measured separately depending on the type of presynaptic neurons and connections to all types of postsynaptic targets were taken into account. Neurons which do not have any outgoing connections in both of the datasets being compared were excluded from the analysis. In Fig. 6e, the number of common connections for adult vs. adult was computed by averaging

the values in three possible pairs (adult-1 vs. adult-2, adult-1 vs. adult-3, adult-2 vs. adult-3) and the number of common connections for dauer vs. adult was computed by averaging the values in three possible pairs (dauer vs. adult-1, vs. adult-2, vs. adult-3).

### Connection similarity

Connection similarity (CS) measures the amount of similarity between two connectivity vectors of a neuron. CS was defined as

$$CS(v_{s1}^i, v_{s2}^i) = exp(-d(v_{s1}^i, v_{s2}^i)) \tag{2}$$

where $d(v_{s1}^i, v_{s2}^i)$ is a Euclidean distance between two connectivity vectors of neuron $i$ (i.e., binary vector of outgoing connections) from stages $s1$ and $s2$. Exponential to the negative power is applied so vectors with greater distances can have lower similarity values and to force the range of the measure between 0 and 1. As the number of common connections, connection similarity has been measured separately depending on the type of presynaptic neurons. Neurons which do not have any outgoing connections in both of the datasets being compared were excluded from the analysis. In Fig. 6f, connection similarity for adult vs. adult was computed by averaging the values in three possible pairs of adult datasets (adult-1 vs. adult-2, adult-1 vs. adult-3, adult-2 vs. adult-3) and connection similarity for dauer vs. adult was computed by averaging the values in three possible pairs of dauer and adult datasets (dauer vs. adult-1, vs. adult-2, vs. adult-3).

### Motif analysis

The whole network has been divided into three subnetworks: sensory, interneuron, and motor subnetworks. Then, the analysis has been done separately for each subnetwork, which only consists of connections within the neurons of that type. To compute the occurrences of three-cell motifs, all combinations of three cells were extracted from the binarized connectivity matrix. For each combination, it was assigned to one of 16 motif patterns depending on its connectivity. The proportion of triangular motifs was computed by dividing the sum of numbers of motifs that form triangles (motifs in black rectangle in Fig. 6b) by the total number of three-cell combinations among 180 neurons. Same computation has been done for 1000 ER random networks, which have the same number of nodes and connection probability, for comparison.

### Vulnerability, Module, and hub detection

In order to assess the vulnerability of the chemical synapse network in both adults and dauer, a systematic elimination of edges was conducted in a random sequence, one at a time. The total number of connected components and the count of nodes in the largest connected component were measured during this process[22]. This procedure was repeated 100 times and averaged. For module detection in chemical synapse networks, the initial 180 ×180 matrices were merged into 97 ×97 matrices, considering left-right pairs. Subsequently, the Clauset-Newman-Moore greedy modularity maximization algorithm[23] was employed to extract the modules. Single-neuron modules were excluded from the results for analytical purposes, yielding 4 modules in each network. The similarity between modules was quantified using the ratio of the *intersection(v1,v2)* to the *union(v1, v2)*, where *v1* and *v2* represent sets of neurons in a module. Hub neurons were identified based on the out-degree of neurons. An Erdős–Rényi (ER) random network was generated 100 times, and the mean of the maximum out-degree for each ER network was utilized as the threshold. Neurons with an out-degree exceeding this threshold were classified as hub neurons[26].

### Statistics

Statistical tests were done by custom Python script using SciPy library. Unpaired t-tests were used to compare the results of behavior experiments (Fig. 3c–e). Data point for dauer was excluded when measuring correlation since it was difficult to decide the proper position for it. Two-sided Wilcoxon rank-sum tests were used to compare distributions of connection weights (Supplementary Fig. 3g) and network properties (Fig. 6).

### Reporting summary

Further information on research design is available in the Nature Portfolio Reporting Summary linked to this article.

## Data availability

The EM image data, volumetric cell segmentation of dauer have been deposited in BossDB and can be downloaded and viewed at https://bossdb.org/project/yim_choe_bae2023 (https://doi.org/10.60533/BOSS-2023-RTPH). Synaptic connectivity data and contact data for dauer are provided as supplementary datas. The EM image data, volumetric cell segmentation, and connectivity data for other developmental stages are publicly available at https://bossdb.org/project/witvliet2020. Source data are provided with this paper.

## Code availability

All analysis code is publicly available at https://github.com/jabae/Yim-Choe-Bae-et_al-2024 (https://doi.org/10.5281/zenodo.10516067). Code for automated synapse detection is available at https://github.com/jabae/Cnapse (https://doi.org/10.5281/zenodo.10516075).

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

## Acknowledgements

We thank Steven J. Cook and Oliver Hobert for reagents and valuable advice. We also thank Scott W. Emmons and Hannes E. Bülow for discussions. *C. elegans* strains were kindly provided by the Caenorhabditis Genetic Center (USA), Yuichi Iino, Elissa A. Hallem. We acknowledge WormAtlas for their valuable resources and for providing reference illustrations. We thank Gwanho Ko and undergraduate students for their help with cell segmentation. We thank Sangwon Son and Daisy S. Lim for helping to make strains. This research was funded by the Samsung Science and Technology Foundation (Project Number SSTF-BA1501-52). H.Yim and D.T.C. were supported by the BK21 program. J.A.B. acknowledges support by the National Research Foundation of Korea (NRF) grant (2019R1A6A1A10073437) funded by the Korean Ministry of Education. K.C.Q.N and D.H.H were funded by NIH OD010943. J.S.K. is supported by the Basic Science Research Programs (2022M3E5E8017946) through the NRF funded by the Korean Ministry of Science and ICT.

## Author contributions

K.N. generated EM dataset after sample preparation by H.K. S.B. stitched and aligned the images. H.K. segmented the EM images and H.Yim proofread them. J.A.B. trained and applied nets for synapses using ground truth generated by H.Yim, D.T.C. H.Yim analyzed morphology and connectivity differences in dauer with help from D.T.C and J.A.B. D.T.C analyzed network characteristics with input from J.A.B. and H.Yim. S.A. M.C., and H.Yim performed confocal experiments. M.C., H.Yang, and H.Yim performed nictation experiments. D.H. supervised image collection. H.Yim, D.T.C., J.A.B., D.H., J.S.K., and J.L. wrote the paper. J.L. and J.S.K. managed the project.

## Competing interests

The authors declare no competing interests.
