## [Peer Review File · Nature Communications]

Comparative connectomics of dauer reveals developmental plasticityREVIEWER COMMENTS

Reviewer #1 (Remarks to the Author):

In this manuscript Yim et al reconstruct the connectome of the nerve ring of the dauer larva of *C. elegans* by combining serial section electron microscopy, volumetric reconstruction, and automated synapse detection using deep learning. The dauer larva is an alternative developmental stage formed in response to adverse environmental conditions. Dauers are developmentally arrested and stress resistant; they have altered morphology, metabolism and behavior, and live for months, but can recover and resume reproductive development in response to favorable environmental conditions.

A connectome is not yet available for the dauer, and is particularly interesting because it would highlight how harsh conditions can rewire a nervous system. Dauer larvae also display specific behaviors, notably nictation.

The authors reconstruct the nerve ring of a dauer larva from 364 serial TEM. They compare their reconstruction to reconstructions made by previous connectomics efforts for other *C. elegans* stages. They highlight dauer specific changes in neuron structure e.g. branching, protrusions and retraction. They describe synapses that are common across stages, as well as dauer specific synapses. Notably, they find a high proportion of dauer-specific connections, and show these are associated with the morphological changes in neuron structure.

General comments

Like other *C. elegans* connectome papers, this work is likely to become a general reference for the community. My main concern is that the nerve ring of only 1 animal was reconstructed. This makes comparisons correspondingly tricky. In fairness, this is a typical complaint of reviewers of connectomics papers, even more so when the paper is studying plasticity, as is the case here. Ideally, the authors should increase the n number to 2, to gain some insight into variability across individuals, but I appreciate this involves substantial effort.

As alternatives: The authors could compare the connectivity of left – right homologous neuron pairs onto left-right homologous targets in the nerve ring of the dauer reconstruction, to assess natural variation in connectivity; they already have these data tabulated. They could also use higher throughput fluorescence microscopy to buttress their EM data. In Fig 1b – e the authors begin to go in this direction, but only provide one image and do not attempt to quantify variation in structure. In fact in the last Discussion section, ‘Limitations’, the authors state that ‘We have conducted fluorescence imaging to validate certain features also exist in other dauer animals but could not validate all features’. The authors

should add some or all of these data to supplementary data. I suggest they chose a set of features and use light microscopy to test their findings. These could include morphological changes in neuron anatomy (e.g. branching, protrusions), and altered synaptic connectivity, which could be examined using iBLINC or GRASP.

Specific comments

1. To reach the broadest audience, the authors should carefully define terms they use, especially in the network analysis. Abbreviations should be used sparingly, and defined, for similar reasons.
2. In places the authors over-sell their work. The paper focuses on the nerve ring of the dauer, but the authors indicate they have a complete connectome (e.g. Abstract, sentence 2: '...we reconstructed the complete connectome of dauer'. This detracts from the manuscript and should be avoided. Electrical connections are also absent in their connectome.
3. The Introduction is short. Providing more information on what distinguishes dauer larvae from other stages, physiologically, anatomically, and behaviourally would help a non-specialist audience. Nictation is analysed in the results section, but is not introduced.
4. The flow of the manuscript could be improved. For example, Lines 78 – 88, on IL2 neurons may be better placed together with the studies of nictation.
5. The authors claim high accuracy because human experts removed false positives. Is there a way to estimate false negatives?

Minor comments

- Throughout the manuscript opened parenthesis are not closed – missing).
- Explain abbreviations Line 69: TEM; line 196: Line 200: M -> M; Line 219: CCs of neurons
- Line 196: Define subnetwork here.
- Line 145 The ASG -> AIA connection stood out

The supplementary data figures are a great addition to the paper.

Reviewer #2 (Remarks to the Author):

Yim et al describe at the ultrastructural level the connectome of a *C. elegans* dauer, which is an alternative 3rd larval stage. They compare their dauer nerve ring connectome to the connectomes of other *C. elegans* stages to demonstrate significant plasticity in the connectome.

This work is inherently descriptive and therefore lacks a single over-arching key punchline or punchlines. However, their results provide a very solid foundation for the formulation of hypotheses regarding circuits of behavior in the dauer. The authors perform one such experiment themselves: they hypothesize based on their EM connectomic data showing a dauer-specific connection between IL2 neurons and RIG neurons, that the latter neuron is involved in dauer nictation behavior. They then perform an optogenetic stimulation experiment that support this hypothesis. They also compare their dauer connectome to adult stage connectome to explain prior observations on the behavioral response to CO₂ in dauers versus adults.

Overall, the EM data presented are of high quality and the analytical tools they use are state of the art. I have only minor critiques of the manuscript, mostly related to the writing. I list a few examples below but would suggest the authors spend time editing their manuscript carefully for readability.

1. Line 145: change “..standed out” to “...stood out...”
2. LINES 271-2: i find it hard to see how reference 20 (on Piezo mechanotransduction) is relevant to this statement that remodeling is a highly energy-consuming process.
3. LINES 377-378 (fig 2 legend) should be “Proportion of stage-specific connections within each dataset. Over a quarter of connections are dauer-specific connections, which is considerably higher than other stages.”
4. LINES 383-384: edit to “Connections with increased or decreased weight in dauer are represented with red or blue arrows, respectively.”
5. LINE 376: edit the sentence to “Area of each region is not scaled to the number of connections.”
6. LINE 179: please provide reference for the graph theoretical analysis method.
7. LINE 219: Define CC (Clustering Coefficient) at first use and provide reference for this method.
8. LINE 257: edit to “...creating differences from adult connectome”
9. LINES 279-281: sentence structure is awkward. May I suggest editing to “Our data sheds light on the observation that in the dauer, unlike in the adult, AIY is not required for CO₂ response. Our data shows that the BAGAIY connection is absent in the dauer.”
10. METHOD lines 478-482: please provide references for each of the strains used in this study.
11. LINE 482: edit to “...*C. elegans* Bristol strain N2 was used.”

12. LINE 534: Change “First of all,...” to “First,...”

Reviewer #3 (Remarks to the Author):

The authors constructed the connectome of a dauer and analyzed the difference between the dauer and the adults. They found that the number of neuronal connections of the dauer is comparable to adults but with a large number of dauer-specific connections. Two neurons involved in the specific connection are experimentally verified to be associated with nictation. The authors also performed graph analysis and discovered significant rewiring in the sensory subnetwork and increased clustering in the motor subnetwork.

This study is particularly interesting and novel for two reasons: 1. dauer is an alternative developmental stage that helps worms increase survivability in harsh conditions. The authors show that the numbers of neurons and connections of a dauer are comparable to adults and are significantly more than other developmental stages. This result suggested the importance of neural network development in survivability despite more energy required to maintain a larger network. 2. Although having comparable numbers of neurons and connections with adults, the authors showed that dauer’s neural network wires differently from adults, and some of the differences may be attributed to the dauer-specific behavior. The study showcases how a neural network rewires and modifies itself to produce specific behavior and survive. The paper is interesting to the general audience of Nature Communications and is potentially suitable for publication in the journal, provided the following comments are addressed:

Major comments:

1. When constructing the connectome, the authors assumed that the size of an active zone is proportional to the synaptic strength. Although this assumption looks intuitive, is there any experimental support for this assumption?
2. the author mainly compared a dauer to the adults for the sensory, interneuron, and motor subnetworks in their graph analysis. It would be very informative if the authors also analyzed the connectivity changes between the subnetworks. Can authors find any changes in the connections between subnetworks (S-I, I-M and S-M)?
3. Following the comment above, an important question to investigate is how the sensorimotor information pathway is changed in the dauer. Nictation must be triggered by sensory cues. So, it is interesting to see whether the development of dauer creates specialized sensorimotor circuits. The authors showed connectivity changes involving IL2Q and ASG (figure 4a) and changes in the sensory subnetwork (figure 6g) involving IL2 neurons. The authors should take a step further by analyzing the circuits involving IL2 and ASG under the context of the sensorimotor pathway. Specifically, in addition to

IL2/RIG/ASG neurons and their connections, can the authors identify complete sensorimotor circuits potentially responsible for nictation?

4. Using the U-net model to identify neuronal connections is a crucial step in constructing the connection in the study. Therefore, the authors need to show how well the U-net model worked. Its performance determines the reliability of the connectome data. The authors should report their U-net model's performance, e.g., the confusion matrix.

5. The paper focuses on the nictation behavior, but a dauer also exhibits other behaviors that differ from adult or other developmental stages. For example, dauers tend to remain motionless. Is this behavioral tendency associated with more inhibition or less dis-inhibition on the motor neurons? Can the authors identify such changes in their connectome data?

6. The functional experiments of RIG and ASG could have been more thoroughly carried out. Can the authors inhibit RIG to show that the nictation ratio decreases? Similarly, how about exciting ASG to show the increasing nictation ratio? If this cannot be done in a reasonable time frame due to the limitation of genetic tools or other reasons, please explain it in the Discussion.

7. I am surprised that the authors did not perform some of the standard analyses that other network studies typically did. For example, community/modularity, hubs, vulnerability, etc. These are the essential properties that help us understand the structure of a network. It would be great to know how these properties change in the dauer connectome versus adult ones.

Minor comments:

1. Line 240-242: "The fact that they have fewer common connections ..." It is unclear what the authors meant here. Please elaborate.

2. Line 262: "As the last piece of the puzzle, ..." ☒ "To solve the last piece of the puzzle, ..."

3. Line 308-309: "However, our classification of interneurons are different from interneurons defined in 5 as it also includes modulatory neurons, which have many variable connections." What does "it" in the sentence refer to? The authors' classification or the classification of Witvliet et al. 2021? According to English grammar, it refers to the latter. But based on the context, it looks like the former. Please clarify.

4. Figure 6h. This plot does not deliver much information and lacks explanation in the legend. I fail to see how those bold arrows in the S network represent the concept of "rewiring" in the sensory subnetwork. Moreover, what do the curved arrows pointing from S to I and M mean? I suggest removing this panel or replacing it with a plot that displays more detailed information.

5. Line 554-559. Use past tense instead of present perfect tense.

6. This study uses three adult datasets. However, in some plots, e.g., Fig 1 d-e middle, Fig 2d, Fig 3c & f, Fig 4a, and Fig 6d & g, it looks like only a single adult is shown or compared. Please specify which adult dataset is used. Or if some plots are based on the average from multiple datasets, please also specify.

RESPONSES TO REVIEWER COMMENTS

Reviewer #1 (Remarks to the Author):

General comments

Like other *C. elegans* connectome papers, this work is likely to become a general reference for the community. My main concern is that the nerve ring of only 1 animal was reconstructed. This makes comparisons correspondingly tricky. In fairness, this is a typical complaint of reviewers of connectomics papers, even more so when the paper is studying plasticity, as is the case here. Ideally, the authors should increase the n number to 2, to gain some insight into variability across individuals, but I appreciate this involves substantial effort.

As alternatives: The authors could compare the connectivity of left – right homologous neuron pairs onto left-right homologous targets in the nerve ring of the dauer reconstruction, to assess natural variation in connectivity; they already have these data tabulated. They could also use higher throughput fluorescence microscopy to buttress their EM data. In Fig 1b – e the authors begin to go in this direction, but only provide one image and do not attempt to quantify variation in structure. In fact in the last Discussion section, ‘Limitations’, the authors state that ‘We have conducted fluorescence imaging to validate certain features also exist in other dauer animals but could not validate all features’. The authors should add some or all of these data to supplementary data. I suggest they chose a set of features and use light microscopy to test their findings. These could include morphological changes in neuron anatomy (e.g. branching, protrusions), and altered synaptic connectivity, which could be examined using iBLINC or GRASP.

- **Response:** Thank you for the valuable comments. We fully agree with your concerns and have revised the manuscript following your suggestions as follows.

< Left-right homologue >

- We added **Supplementary Fig.3e,f** for this. Supplementary Fig 3e is a graph, similar to Fig. 5a of a previous connectome paper (Cook et al., 2019), showing the weight of left-right homologous neuron pairs in dauer. Synapse weights are normalized values. The result of linear regressing them is $y=0.89x+0.16$, which is close to $y=x$, and the coefficient of determination (R^2) is 0.648, which shows a strong left-right correlation. Taken together, our dauer dataset shows a high left-right similarity.

Supplementary Fig. 3f is an application of the connection similarity (CS) used in Fig. 6f of our manuscript. Given a pair of neurons that are left/right homologues, each neuron's partner connection could be represented as a binary vector. Connection similarity is a measure of how similar these vectors are. Each left/right neuron pair has one datapoint, and the higher the value of the datapoint means the higher the L/R partner similarity. The result is shown in the boxplot for all stages. In early stages such as L1, the CS is high because there are fewer variable synapses, and the average CS decreases as the maturation of stages. In the case of dauer, it is fair to compare with adult because it has many properties similar to adult such as the number of connections. The CS of dauer shows a similar degree when compared with adult datasets, or even better, therefore it shows that the left - right pair of dauer is quite similar to those of other adult datasets.

Revised text: (line 113-115) **The left-right homologous neuron pairs in the dauer show strong correlation, and homologous connection shows similar connection similarity with adults (Supplementary Fig. 3e,f; Methods).**

Supplementary Fig.3e Correlations between left and right homologous connections weight(normalized) in dauer. $R^2 = 0.648$. (Shaded areas around the regression line: 95% confidence intervals. Two-sided t-test; $p=0.000$)

Supplementary Fig.3f Connection similarity of left/right neuron pairs from each dataset (Wilcoxon rank-sum test; $**p<0.01$). The results showed that the left/right similarity in dauer is not significantly different from, or even better than, that in adults.

<Validation of neuron morphology>

- We tried to identify other features by fluorescence microscopy, but many of the changes seems too subtle to be seen with our x1000 confocal. However, we were able to confirm the protrusion of the IL2 quadrants using the *kfp-6* promoter, which is shown in new Fig. 1b, c. We wanted to further confirm big changes such as branching, but a single promoter was not available for any of the candidate neurons. We tried to identify new promoters using CENGEN data, but had difficulty in identifying them due to overlapping expression between multiple neurons, so it was practically impossible to further confirm them with fluorescence during our 3-month-period for revision.

I would like to try to convince the reviewer about the reason why we were confident of dauer-specific morphology even with $n= 1$: it is because we have another set of unpublished dauer EM images (in preparation); we are preparing a paper on the individuality of the nervous system in which we compare connectomes, both chemical and electrical, of two individual dauers. In addition, this dauer dataset is rather short and tells us less about the wiring of the full nerve ring, so we did not include this dataset in the current paper. But even in this EM set, we found that the dauer-specific morphological changes were well reproduced, and we are confident that these changes are not the results of the individuality of the EM dataset. For example, the following are some examples of the reproducible changes in the dauer neurons (both dauer and dauer-short sets). Although we have not included the data in the revision, we would be willing to include them if the editor and reviewer recommend so.

As for quantification, we performed fluorescence quantification on IL2 lateral neurons because they show the biggest and clearest neuronal morphology changes in fluorescence. The branching of IL2 lateral neurons was checked for L4, dauer, and post-dauer L4, with $n=20$ for each stage. We found that 0/20 of L4, 20/20 of dauer, and 1/20 of post-dauer L4 (one animal might not yet fully recovered from

dauer) shows branching, indicating that dauer-specific morphological change is a very strong binary trait. We added this result in the revision.

Revised text:

(line 84-97) Dauer IL2 quadrant neurons add an extra protrusion at the axon bend (Fig 1b, c). The dendrite arborizations of IL2 quadrant neurons were well-reported, but any structural change in axon was not reported before¹⁷. Thanks to the finer resolution of TEM, we discovered that remodeling occurs in axons of IL2 neurons as well. As a validation, we employed fluorescence microscopy to observe the IL2 quadrant neurons in different stages including a post-dauer adult (Fig. 1c), confirming that the extra protrusions of IL2 quadrant neurons appear only in dauer.

Dauer IL2 lateral neurons also show distinctive structural changes in dauer, add an extra branch that emerges perpendicularly from the apex of an axon bend (Fig. 1d,e). These changes were unreported before. We performed fluorescence quantification on IL2 lateral neurons because they showed the biggest and clearest neuronal morphology changes in fluorescence. The branching of IL2 lateral neurons was checked for L4, dauer, and post-dauer L4, with n=20 for each stage. We found that 0/20 of L4, 20/20 of dauer, and 1/20 of post-dauer L4 showed branching, indicating that dauer-specific morphological change is a very strong binary trait. In addition, the terminal swellings of IL2 lateral neurons were shrunken in dauer, losing contact with the IL2 dorsal neurons (Fig. 1d).

Fig. 1b Volumetric reconstruction of IL2DR (quadrant) neurons in L3, adult-2, dauer (from left) and schematic view of neuron in dauer stage (right). New protrusion emerges from the axon (yellow arrow, solid yellow).

Fig. 1c Fluorescence images of IL2 quadrant neurons in L3, adult, dauer, and post-dauer adult (from left). IL2 lateral neurons were marked with GFP using the *kfp-6* promoter.

<Validation of synapse>

For synapse validation, we performed GRASP on IL2-RIG neuron connections and supplemented Fig. 4b. In other normal developmental stages and post-dauer stages, we did not see any GRASP signal, but we could observe dauer-specific GFP signal. This is a good example that dauer-specific synaptic changes are not only found in EM but also visible in live animals.

Revised text:

(line 168-170) In dauer, each IL2 process forms a new swelling, forming new contact and large synapse with RIG neurons(Fig. 4b), and synapses were confirmed well by GRASP(Fig.4c)¹⁹.

Fig. 4b Fluorescence images of IL2 neurons and GRASP signal in adult (top) and dauer (bottom) stages. IL2 neurons were marked with mCherry using f28a12.3 promoter. Splited GFP11 were tagged with IL2 neurons using f28a12.3 promoter, and Splited GFP1-10 were tagged with RIG neurons using twk-3 promoter.

Specific comments

1. To reach the broadest audience, the authors should carefully define terms they use, especially in the network analysis. Abbreviations should be used sparingly, and defined, for similar reasons.

- **Response:** Thank you for pointing this out. We checked and explained abbreviations that we used, specially which you mentioned in minor comment.

2. In places the authors over-sell their work. The paper focuses on the nerve ring of the dauer, but the authors indicate they have a complete connectome (e.g. Abstract, sentence 2: ‘...we reconstructed the complete connectome of dauer’. This detracts from the manuscript and should be avoided. Electrical connections are also absent in their connectome.

- **Response:** Thank you for pointing this out. We revised the manuscript as follows:

Revised text: (line 22) To address this, we reconstructed the **chemical** connectome of dauer....

3. The Introduction is short. Providing more information on what distinguishes dauer larvae from other stages, physiologically, anatomically, and behaviourally would help a non-specialist audience. Nictation is analysed in the results section, but is not introduced.

- **Response:** Thank you for pointing this out. We revised the manuscript as follows:

Added text: (line 54-57) **Dauers also show a stage-specific hitchhiking behavior, nictation, in which dauers stand with their tails and wave the bodies to facilitate phoresy to other carrier animals. We previously showed that IL2 neurons are responsible for the nictation behavior, but it is not well understood why only dauers, but not other stage animals, are capable of the nictation behavior¹²**

4. The flow of the manuscript could be improved. For example, Lines 78 – 88, on IL2 neurons may be better placed together with the studies of nictation.

- **Response:** Thank you for the suggestion. Because we added explanations on nictation studies in Introduction, this result part may read smoothly now. We also added results of new structural changes in IL2Q neurons here. The revised manuscript is as follows:

Revised and added text:

(line 84-97) **Dauer IL2 quadrant neurons add an extra protrusion at the axon bend (Fig 1b, c). The dendrite arborization of IL2 quadrant neurons were well-reported, but any structural change in axon was not reported before¹⁷. Thanks to the finer resolution of TEM, we discovered that remodeling occurs in axons of IL2 neurons as well. As a validation, we employed fluorescence microscopy to observe the IL2 quadrant neurons in different stages including a post-dauer adult (Fig. 1c), confirming that the extra protrusion of IL2 quadrant neurons appear only in dauer.**

Dauer IL2 lateral neurons also show distinctive structural changes in dauer, add an extra branch that emerges perpendicularly from the apex of an axon bend (Fig. 1d,e). These changes were unreported before. We performed fluorescence quantification on IL2 lateral neurons because they showed the biggest and clearest neuronal morphology changes in fluorescence. The branching of IL2 lateral neurons was checked for L4, dauer, and post-dauer L4, with n=20 for each stage. We found that 0/20 of L4, 20/20 of dauer, and 1/20 of post-dauer L4 showed branching, indicating that dauer-specific morphological change is a very strong binary trait. In addition, the terminal swellings of IL2 lateral neurons were shrunken in dauer, losing contact with the IL2 dorsal neurons (Fig. 1d).

5. The authors claim high accuracy because human experts removed false positives. Is there a way to estimate false negatives?

- **Response:** Thank you for bringing this up. We agree it is an important point to consider. In order to evaluate the overall performance of the synapse detection, we chose two randomly-selected independent cutouts as test volumes to conduct evaluation. These volumes were not overlapping with the regions we have used for the training data.

There are two parts to the synapse detection pipeline: 1) active zone prediction and 2) synaptic partner

assignment. The second part, the partner assignment, is a deterministic process meaning once the active zone is detected properly, the result will not change. Therefore, we conducted evaluation on the first part, the active zone prediction.

To generate the ground truth for the test volumes, two annotators have independently labeled the active zones in the test volumes. Then, the both annotators discussed with third-person (reviewer) to finalize the ground truth for the test volumes. For the evaluation, we evaluated whether each active zone segment has been predicted properly or not. If the predicted active zone is overlapping with labeled segment in the ground truth, that would be a true positive. And if the predicted active zone does not have any overlap with any of the segment in the ground truth, it would be a false positive and false negative if vice versa. As a result, we were able to get over 90% precision and 72% recall. As we have claimed, we have high precision as we proofread all the detected active zones manually to remove false positives. According to our evaluation, it is true that we are missing more active zones than the falsely predicted ones. However, we believe it does not affect subsequent analyses for the following reasons. First of all, most of our analyses are based on connectivity (pairs with at least one synapse are defined to be connected) so the error in connectivity is expected to be smaller even with occasionally missed predictions. Next, we focus on connections that are added or increased in strength in the manuscript (e.g., IL2->RIG connection). As our synapse detection pipeline has very high precision, we are certain that these findings are valid.

Automated synapse detection methods may not be perfect but it helps significantly to avoid biased predictions, which often could cause manual annotation to have a number of false positives. Besides, there are many more vague synapses observed in invertebrates. Therefore, there could be a slight decline in the performance with limited distribution ground truth used for training as observed in the automated synapse detection for *drosophila* whole-brain reconstruction (Dorkenwald et al. bioRxiv. 2023; Buhmann et al. Nature Methods. 2021). Compared to this study, our numbers are in fact not low as we have higher recall given the same precision. Since we have initially reconstructed this dataset, we have improved the automated synapse detection pipeline so we are also planning to publish the improved version of connectivity data online upon submission.

Buhmann, Julia, Arlo Sheridan, Caroline Malin-Mayor, Philipp Schlegel, Stephan Gerhard, Tom Kazimiers, Renate Krause, et al. 2021. "Automatic Detection of Synaptic Partners in a Whole-Brain Drosophila Electron Microscopy Data Set." *Nature Methods* 18 (7): 771–74.

Dorkenwald, Sven, Arie Matsliah, Amy R. Sterling, Philipp Schlegel, Szi-Chieh Yu, Claire E. McKellar, Albert Lin, et al. 2023. "Neuronal Wiring Diagram of an Adult Brain." *bioRxiv : The Preprint Server for Biology*, July. <https://doi.org/10.1101/2023.06.27.546656>.

We added the analysis data as Supplementary Fig. 1d, and the analysis method to Methods.

Revised text: Supplementary Fig. 1d Confusion matrix of synapse detection model. The numbers indicate the number of active zones. Since the active zone segments cannot be defined in unpredicted regions, true negative cannot be quantified.

Revised Methods:

(line 614–623) Evaluation of synapse detection model

Two test volumes of size 1024 x 1024 x 1600 nm³ have been randomly selected from regions that were not covered in the training data. To generate the ground truth for the test volumes, two annotators have independently labeled the active zones in the volumes. Then, both annotators and the third annotator reviewed the volume and proofread the annotations.

We compared the predicted active zone segments and counted predicted active zone segments that overlap with ground truth segments (true positive), predicted segments that do not overlap with any ground truth segment (false positive), and ground truth segments that do not overlap with any predicted segment (false negative). Since we are not able to define active zone segments in the undetected region, true negatives could not be quantified.

Minor comments

- Throughout the manuscript opened parenthesis are not closed – missing).

- **Response:** Thank you for pointing this out. We checked whole manuscripts, and found that we missed some parenthesis ex) (Fig. 6a; ⁴. We revised it.

- Explain abbreviations Line 69: TEM; line 196: Line 200: M -> M; Line 219: CCs of neurons

- **Response:** Thank you for pointing this out. For abbreviation S, I, M, we explain them in Fig.6a legend. CC was explained in line 190.

Revised text: (line 72) *A. C. elegans* dauer nerve ring sample was sectioned and imaged at every 50 nm, resulting in 364 serial **transmission electron microscopy**(TEM) images. (Fig.6a legend for M->M) **S: sensory, I : inter, M : motor neurons**. (line 200-202) As dauer and adult stages consist of more connections, it leads to shorter mean path length (Fig. 5c) and higher **clustering coefficient (CC)**, which measures the probability of the pair of neurons that are connected to a common neuron being connected to each other (Fig. 5d)

- Line 196: Define subnetwork here.

- **Response:** Thank you for your comments.

Revised text: (line 208-211) While the dauer and adult stages share common features in primary network properties, we further explored whether there are differences between dauer and adult stages in subnetwork(**a subset of the whole network with pre- and postsynaptic neurons as one of sensory, inter-, or motor neurons**) properties.

- Line 145 The ASG -> AIA connection stood out

- **Response:** Thank you for pointing this out.

Revised text: (line 153-155) The ASG→AIA connection **stood** out among the 31 dauer-loss connections, as it is a strong connection in other developmental stages which has totally disappeared in dauer (Fig. 3b,c; Supplementary Table 4).

- The supplementary data figures are a great addition to the paper.

- **Response:** Thank you very much for your compliment. We hope this material is of help to other *C. elegans* neuroscientists, intuitively.

Reviewer #2

1. Line 145: change “..standed out” to “...stood out...”

- **Response:** Thank you for pointing this out.

Revised text: (line 153-155) The ASG→AIA connection **stood** out among the 31 dauer-loss connections, as it is a strong connection in other developmental stages which has totally disappeared in dauer (Fig. 3b,c; Supplementary Table 4).

2. LINES 271-2: i find it hard to see how reference 20 (on Piezo mechanotransduction) is relevant to this statement that remodeling is a highly energy-consuming process.

- **Response:** Thank you for pointing this out. We found this reference is less relevant, so we removed it and added new reference: DeWane, G., Salvi, A. M. & DeMali, K. A. Fueling the cytoskeleton - links between cell metabolism and actin remodeling. J. Cell Sci. 134, (2021).

3. LINES 377-378 (fig 2 legend) should be “Proportion of stage-specific connections within each dataset. Over a quarter of connections are dauer-specific connections, which is considerably higher than other stages.”

- **Response:** Thank you for pointing this out.

Original text: Over a quarter of connections are dauer-specific connections in dauer, which are considerably higher than other stages.

Revised text: (line 402-403) **Over a quarter of connections are dauer-specific connections, which are considerably higher than other stages.**

4. LINES 383-384: edit to “Connections with increased or decreased weight in dauer are represented with red or blue arrows, respectively.”

- **Response:** Thank you for pointing this out.

Original text: Connections which connection weight increased and decreased in dauer are represented with red and blue arrows respectively.

Revised text: (line 408-409) **Connections with increased or decreased weight in dauer are represented with red or blue arrow, respectively.**

5. LINE 376: edit the sentence to “Area of each region is not scaled to the number of connections.”

- **Response:** Thank you for pointing this out.

Original text: Area of each region is not scaled according to the number of connections for visual purposes.

Revised text: (line 401) **Area of each region is not scaled to the number of connections.**

6. LINE 179: please provide reference for the graph theoretical analysis method.

- **Response:** Thank you for pointing this out. Also, other network features are included in Supplementary Fig. 6, so we added some new references:

Dijkstra, E. W. A note on two problems in connexion with graphs. *Numer. Math.* 1, 269–271 (1959).

Holme, P., Kim, B. J., Yoon, C. N. & Han, S. K. Attack vulnerability of complex networks. *Phys. Rev. E Stat. Nonlin. Soft Matter Phys.* 65, 056109 (2002)

Clauset, A., Newman, M. E. J. & Moore, C. Finding community structure in very large networks. *Phys. Rev. E Stat. Nonlin. Soft Matter Phys.* 70, 066111 (2004)

Hubert, L. & Arabie, P. Comparing partitions. *J. Classification* 2, 193–218 (1985)

Barrat, A., Barthélemy, M., Pastor-Satorras, R. & Vespignani, A. The architecture of complex weighted networks. *Proc. Natl. Acad. Sci. U. S. A.* 101, 3747–3752 (2004)

van den Heuvel, M. P. & Sporns, O. Network hubs in the human brain. *Trends Cogn. Sci.* 17, 683–696 (2013).

7. LINE 219: Define CC (Clustering Coefficient) at first use and provide reference for this method.

- **Response:** Thank you for pointing this out. We added new reference:

Fagiolo, G. Clustering in complex directed networks. *Phys. Rev. E Stat. Nonlin. Soft Matter Phys.* 76, 026107 (2007).

8. LINE 257: edit to "...creating differences from adult connectome"

- **Response:** Thank you for pointing this out.

Original text: In conclusion for the graph theoretical analyses, there exists increased clustering among motor neurons and significant rewiring in the connectivity of sensory neurons in the dauer connectome, creating differences with adult connectomes.

Revised text: (line 269-271) In conclusion for the graph theoretical analyses, there exists increased clustering among motor neurons and significant rewiring in the connectivity of sensory neurons in the dauer connectome, creating differences **from** adult connectomes.

9. LINES 279-281: sentence structure is awkward. May I suggest editing to “Our data sheds light on the observation that in the dauer, unlike in the adult, AIY is not required for CO₂ response. Our data shows that the BAGAIY connection is absent in the dauer.”

- **Response:** Thank you for your nice suggestion.

Original text: Consistent with a previous report that AIY is not required for CO₂ response in dauers unlike adults²¹, we easily identified that the connection from BAG¹¹, which is responsible for CO₂ detection, to AIY does not exist in the dauer connectome while it exists in the adult.

Revised text: (line 293-295) **Our data sheds light on the observation that in the dauer, unlike in the adult, AIY is not required for CO₂ response²¹. Our data shows that the BAG->AIY connection is absent in the dauer¹¹.**

10. METHOD lines 478-482: please provide references for each of the strains used in this study.

- **Response:** Thank you for pointing this out. We added more strains during revision, so we updated whole references.

Lee, H. et al. Nictation, a dispersal behavior of the nematode *Caenorhabditis elegans*, is regulated by IL2 neurons. *Nat. Neurosci.* 15, 107–112 (2011).

Jang, M. S., Toyoshima, Y., Tomioka, M., Kunitomo, H. & Iino, Y. Multiple sensory neurons mediate starvation-dependent aversive navigation in *Caenorhabditis elegans*. *Proc. Natl. Acad. Sci. U. S. A.* 116, 18673–18683 (2019).

Guillermin, M. L., Carrillo, M. A. & Hallem, E. A. A Single Set of Interneurons Drives Opposite Behaviors in *C. elegans*. *Curr. Biol.* 27, 2630–2639.e6 (2017).

11. LINE 482: edit to “...*C. elegans* Bristol strain N2 was used.”

- **Response:** Thank you for pointing this out.

Original text: *C. elegans* strains used in this study are as follows: N2 Bristol, *cha-1(n2411)*, *daf-2(e1370)*, *daf-10(e1387)* lines were obtained from the *Caenorhabditis* Genetics Center (CGC).

Revised text: (line 507-508) *C. elegans* Bristol strain N2, *cha-1(n2411)*, *daf-2(e1370)*, *daf-10(e1387)* lines were used, obtained from the *Caenorhabditis* Genetics Center (CGC)¹⁶.

12. LINE 534: Change “First of all,...” to “First,...”

- **Response:** Thank you for pointing this out.

Original text: First of all, image patches were normalized to have similar contrast and brightness.

Revised text: (line 562-563) **First**, image patches were normalized to have similar contrast and brightness.

Reviewer #3

Major comments:

1. When constructing the connectome, the authors assumed that the size of an active zone is proportional to the synaptic strength. Although this assumption looks intuitive, is there any experimental support for this assumption?

- **Response:** Thank you for the comment. There is evidence suggesting presynaptic active zone area is correlated with synaptic strength. In mammalian neurons, there have been discoveries that show dendritic spine volume is correlated with physiological strength of synapse (Matsuzaki, 2004; Noguchi, 2011; Holler, 2021). Besides, it has been long known that spine volume is correlated with synaptic structure features such as postsynaptic density, number of vesicles, and also presynaptic active zones (Harris, 1989). Based on these studies, many previous studies using EM have been approximating postsynaptic density size or presynaptic active zone size as synaptic strength. Along these lines, since 1986 Mind of a worm paper², *C. elegans* connectome papers also tends to consider synaptic strength under the assumption that the size of an active zone is proportional to the synaptic strength^{4,5,15} as they use the number of sections each active zone spans as the synaptic strength.

Matsuzaki, M., Honkura, N., Ellis-Davies, G. C. R. & Kasai, H. Structural basis of long-term potentiation in single dendritic spines. *Nature* **429**, 761–766 (2004).

Noguchi, J. *et al.* In vivo two-photon uncaging of glutamate revealing the structure-function relationships of dendritic spines in the neocortex of adult mice. *J. Physiol.* **589**, 2447–2457 (2011).

Holler, S., Köstinger, G., Martin, K. A. C., Schuhknecht, G. F. P. & Stratford, K. J. Structure and function of a neocortical synapse. *Nature* **591**, 111–116 (2021).

Harris, K. M. & Stevens, J. K. Dendritic spines of CA 1 pyramidal cells in the rat hippocampus: serial electron microscopy with reference to their biophysical characteristics. *J. Neurosci.* **9**, 2982–2997 (1989).

2. the author mainly compared a dauer to the adults for the sensory, interneuron, and motor subnetworks in their graph analysis. It would be very informative if the authors also analyzed the connectivity changes between the subnetworks. Can authors find any changes in the connections between subnetworks (S-I, I-M and S-M)?

- **Response:** Thank you for your suggestion. In **Supplementary Fig. 7b, 8b, 8d**, we added more features about changes between the subnetworks.

First, we showed the results for out-degree in Fig. 6a, so we showed the results for in-degree in Supp. Fig. 7b. Certain connection types (M>S, M>I) showed significance.

The number of mean connections provided in Supplementary Fig. 8b did not show any significance when analyzed between subnetwork type. As shown in Fig. 6e, there was a difference in sensory neurons when the output results of each type were combined, so it can be seen that in the case of sensory, the

difference in S>S, S>I, S>M accumulates to make a significant difference.

When we look at connection similarity between subnetwork types, we see differences between S>I and M>M. It could be described that S>I similarity drives the difference in similarity across S>, just as the increase in the number of connections in M>M affects connection similarity across M> (especially, since there is no difference in the number of connections in S>I as opposed to M>M). However, we interpreted this result as similarity of whole sensory output(S>), because S>S, S>I, and S>M all made a cumulative significant difference in the number of mean connections. Therefore, we think that the connections outgoing from dauer sensory neurons are different overall, and that dauer process sensory cues differently than adults.

Revised text:

Supplementary Fig. 7b. In-degrees of neurons for different connection types (Wilcoxon rank-sum test; $n=66(M \rightarrow S)$, $n=73(M \rightarrow I)$, $**p < 0.01$) in adults (blue) and dauer (orange). Neurons without any connection in any of the datasets were excluded. S: sensory, I: inter, M: motor neurons.

Supplementary Fig. 8b. Number of common output connections for different connection types between adults (left) and that between dauer and adults (right). Neurons without any connection in both datasets of the pair were excluded. Colors are based on the presynaptic neuronal class (sensory: pink, inter-: orange, motor: blue).

Supplementary Fig. 8d. Connection similarity of output connections for different connection types between adults (left) and that between dauer and adults (right; Wilcoxon rank-sum test; $**p < 0.01$, $***p < 0.001$). Neurons without any connection in both datasets of the pair were excluded. Colors are based on the presynaptic neuronal class (sensory: pink, inter-: orange, motor: blue).

3. Following the comment above, an important question to investigate is how the sensorimotor information pathway is changed in the dauer. Nictation must be triggered by sensory cues. So, it is interesting to see whether the development of dauer creates specialized sensorimotor circuits. The authors showed connectivity changes involving IL2Q and ASG (figure 4a) and changes in the sensory subnetwork (figure 6g) involving IL2 neurons. The authors should take a step further by analyzing the circuits involving IL2 and ASG under the context of the sensorimotor pathway. Specifically, in addition to IL2/RIG/ASG neurons and their connections, can the authors identify complete sensorimotor circuits potentially responsible for nictation?

- **Response:** Thank you for your great suggestion. Our lab has been trying to unlock the secrets of nictation through very different approaches. We think nictation is a very complex behavior, which involves the *daf-2* and *daf-7* pathways, sensorymotor pathway, parallel coordination circuits that presumably involve GLR cells, and the action of head and body muscles. Unfortunately, we only revealed the chemical connectome of nerve ring in dauer, so we barely approached the upstream neurons of nictation circuits, working in the head region. We do not know the whole-body connectome, which is not revealed yet, and also we need information on gap junctions in order to explain real circuits of behavior. Please understand that it is difficult to tell a complete sensorimotor circuit at this point. But we do appreciate your interest in this. It is our ultimate goal to reveal the complete nictation circuit. We added some of our thoughts in Limitations.

4. Using the U-net model to identify neuronal connections is a crucial step in constructing the connection in the study. Therefore, the authors need to show how well the U-net model worked. Its performance determines the reliability of the connectome data. The authors should report their U-net model's performance, e.g., the confusion matrix.

- **Response:** We agree that performance of the synapse detection model is critical and thank you for the constructive feedback. In **Supplementary Fig 1d**, we have added a confusion matrix evaluating the performance of our synapse detection model on test volumes.

In order to evaluate the overall performance of the synapse detection, we chose two randomly-selected independent cutouts as test volumes to conduct evaluation. These volumes were not overlapping with the regions we have used for the training data.

There are two parts to the synapse detection pipeline: 1) active zone prediction and 2) synaptic partner assignment. The second part, the partner assignment, is a deterministic process meaning once the active zone is detected properly, the result will not change. Therefore, we conducted evaluation on the first part, the active zone prediction.

To generate the ground truth for the test volumes, two annotators have independently labeled the active zones in the test volumes. Then, the both annotators discussed with third-person (reviewer) to finalize the ground truth for the test volumes. For the evaluation, we evaluated whether each active zone segment has been predicted properly or not. If the predicted active zone is overlapping with labeled segment in the ground truth, that would be a true positive. And if the predicted active zone does not have any overlap with any of the segment in the ground truth, it would be a false positive and false negative if vice versa. As a result, we were able to get over 90% precision and 72% recall.

As we have claimed, we have high precision as we proofread all the detected active zones manually to remove false positives. According to our evaluation, it is true that we are missing more active zones than the falsely predicted ones. However, we believe it does not affect subsequent analyses for the following reasons. First of all, most of our analyses are based on connectivity (pairs with at least one synapse are defined to be connected) so the error in connectivity is expected to be smaller even with occasionally missed predictions. Next, we focus on connections that are added or increased in strength in the manuscript (e.g., IL2->RIG connection). As our synapse detection pipeline has very high precision, we are certain that these findings are valid.

Automated synapse detection methods may not be perfect but it helps significantly to avoid biased predictions, which often could cause manual annotation to have a number of false positives. Besides, there are many more vague synapses observed in invertebrates. Therefore, there could be a slight decline in the performance with limited distribution ground truth used for training as observed in the automated synapse detection for *drosophila* whole-brain reconstruction (Dorckenwald et al. bioRxiv. 2023; Buhmann et al. Nature Methods. 2021). Compared to this study, our numbers are in fact not low as we have higher recall given the same precision. Since we have initially reconstructed this dataset, we have improved the automated synapse detection pipeline so we are also planning to publish the improved version of connectivity data online upon submission.

Buhmann, Julia, Arlo Sheridan, Caroline Malin-Mayor, Philipp Schlegel, Stephan Gerhard, Tom Kazimiers, Renate Krause, et al. 2021. "Automatic Detection of Synaptic Partners in a Whole-Brain Drosophila Electron Microscopy Data Set." *Nature Methods* 18 (7): 771–74.

Dorckenwald, Sven, Arie Matsliah, Amy R. Sterling, Philipp Schlegel, Szi-Chieh Yu, Claire E. McKellar, Albert Lin, et al. 2023. "Neuronal Wiring Diagram of an Adult Brain." *bioRxiv : The Preprint Server for Biology*, July. <https://doi.org/10.1101/2023.06.27.546656>.

5. The paper focuses on the nictation behavior, but a dauer also exhibits other behaviors that differ from adult or other developmental stages. For example, dauers tend to remain motionless. Is this behavioral tendency associated with more inhibition or less dis-inhibition on the motor neurons? Can the authors identify such changes in their connectome data?

- **Response:** Thank you for your great comment. Dauer-specific changes in CO₂ attraction are a good example, and we have already described our hypothesis in the Discussion. We hypothesize that AIY is not required for CO₂ response in dauer-specific manner. Also, dramatically increased connection between AIB>AVE could affect the CO₂ attractiveness. To help visual understanding, we attached the figure for the CO₂ sensing circuit.

The decrease of pharyngeal pumping in dauer stage is also worth studying. The RIP neurons, which is the only neuron that connects the somatic and pharyngeal connectome, have connections with many neurons and all of them are dramatically changed (increased/decreased) in dauer. Pharyngeal pumping can be further elucidated by studying the pharynx connectome and RIP neurons, but unfortunately there is no dauer-specific pharyngeal connectome yet. However, it is worth further research in this direction for those interested in the pharynx.

We are afraid that being motionless is a physiological state that is difficult to explain at the circuit level, but here, we can give an idea. In Fig. 6a, the I>M out-degree decreases dauer-specifically, and in Supplementary Fig. 6b, the M>S and M>I in-degrees decrease with dauer. Taken together, this suggests that motor neurons are somewhat isolated from sensory and interneurons. The overall isolation of motor neurons in terms of network connectivity can be used as a strategy to conserve energy by rewiring the motor output to not respond except for really important input signals. And for important input signals, the increased clustering between motor neurons allows for faster signal processing. In Supplementary Fig. 6d, the Rand index between type and module shows that dauer is higher than that of adults, which can be seen as a result of increased modularization due to the isolation of motor neurons. All of the above hypotheses will be good topics for future research, and our dauer connectome resource will be very helpful. We therefore revised our manuscript at follows:

Revised text:

(line 299-308) The absence of pharyngeal pumping in dauer stage is also worth studying. The RIP neurons are the only neurons that connect the somatic and pharyngeal connectome and all of their connections are dramatically changed in dauer (Supplementary Data 1). It would be interesting to examine possible roles of RIP neurons in the regulation of pumping of dauer. It is not trivial to discuss the motionless trait of dauer but we could get some hints from the network properties of motor neurons. Motor neurons are somewhat isolated from sensory and interneurons in dauer (Fig. 6a; Supplementary Fig. 6b). A testable hypothesis is that the overall isolation of motor neurons in terms of

network connectivity can be used as a strategy to conserve energy by rewiring the motor output to not responding except for really important input signals.

6. The functional experiments of RIG and ASG could have been more thoroughly carried out. Can the authors inhibit RIG to show that the nictation ratio decreases? Similarly, how about exciting ASG to show the increasing nictation ratio? If this cannot be done in a reasonable time frame due to the limitation of genetic tools or other reasons, please explain it in the Discussion.

- **Response:** Thank you for this constructive suggestion. We also felt that this experiment was a priority. For the RIG ablation experiment, we used the RIG ablation line obtained from a previous paper (Guillermin et al., 2017). The results clearly showed that nictation was significantly decreased when RIG was ablated. We added the results in the revision.

In the case of ASG experiment, the *gcy-21* promoter, which we used for ASG ablation, was too low in expression in dauer. Instead we used *Iron-4* promoter and confirmed its expression in ASG neurons in dauer with mCherry, before proceeding with the experiment. When we activated ASG neurons, nictation was not different with control, but it was not unexpected to us. This result indicates that ASG neurons are a necessary, but not sufficient, condition for nictation. More practically, ASG is not a direct player in the nictation circuit like IL2 neurons or RIG neurons, but it is a necessary neuron for normal nictation behavior. ASG neurons exhibits a Daf-c phenotype upon ablation, and along with ASI neurons, they are representative neurons known to be involved in dauer entry (Bargmann, 1991). Therefore, the absence of ASG suggests that neuronal mechanism during dauer entry itself is altered and furthermore, the physiology of dauer may be affected. the ASG ablation line that we used in the experiment, ASG is ablated at an early stage by caspase. So dauer properties from this line could be different from N2 dauer's. It will be interest to study the crosstalk of ASG-ASI neurons during dauer entry as follow-up research topics.

These results are included in the revision. RIG ablation and ASG ablation/activation results are in Fig.4c and Supplementary Fig.5.

Revised text:

(line 181-182) We found that ASG-ablated worms exhibited deficit in nictation behavior and that **optogenetic activation of ASG did not increase the nictation ratio (Supplementary Fig. 5b,c)**, suggesting that **ASG neurons are necessary, but not sufficient, for nictation behavior** and that **ASG neuron may not be a member of the direct circuit of nictation**.

Fig. 4c RIG ablated lines show significantly lower nictation ratio than N2 (individual nictation test; $n=49$; unpaired t-test; $p<0.0001$).

Fig. 5b ASG ablated lines show significantly lower nictation ratio than N2 (individual nictation test; $n=36$; unpaired t-test; $p=0.0012$).

7. I am surprised that the authors did not perform some of the standard analyses that other network studies typically did. For example, community/modularity, hubs, vulnerability, etc. These are the essential properties that help us understand the structure of a network. It would be great to know how these properties change in the dauer connectome versus adult ones.

- **Response:** Thank you for your suggestion. We added features (community, hub-related properties, vulnerability, etc.) for the interest of the general audience in **Supplementary Fig. 6**. We had to add these results in a Supplementary figure because we did not find significant differences in the global

network structure dimension other than the analyses provided in Fig. 5. In most cases, it was difficult to verify the statistical significance of seemingly differences, as shown in Supplementary Fig. 5c and d. In fact, global properties are most affected by the number of connections when looking at different measures, as shown in Fig. 5 and Supplementary Fig. 6. Therefore, we tend to think that it is more appropriate to look at the subnetwork or specific circuit level to see differences between adults and dauer. We revised the manuscript as suggested by the reviewer as follows.

Revised text: (line 203-204) **The other basic network features are shown in Supplementary Fig. 6²²⁻²⁶.**

Supplementary Fig. 6. Other basic network properties

a Vulnerability of chemical synapse networks across development measured by the number of connected components (top) and the number of nodes in the largest connected component (bottom). Lines are color-coded according to the developmental stages (gray: L1s, light blue: L2 to L4, dark blue: adults, red: dauer).

b Similarity between modules(subset of neurons) among adults (top row) and between dauer and adults (bottom row). Single-neuron modules from the result are excluded. Order of modules are rearranged for visualization purposes so that the pairs with maximum similarity would be positioned on the main diagonal.

c Rand index of modules between adults (blue) and between dauer and adult (orange). The Rand index quantifies the similarity between two partitions, meaning that each matrix in **b** can be condensed into a single value using the Rand index. The annotation above each dot indicates the datasets being compared (A1: Adult-1, A2: Adult-2, A3: Adult-3, Da: Dauer).

d The Rand index between the modular partitioning and the neuronal class (comprising sensory, inter-, and motor neuron) categorization. The neuronal class classifications in ⁴ (top) and ⁵ (bottom) were both used as the basis for assessing the congruence of the partitioned outcomes with the class categorization. (Blue: Adults, Orange: Dauer)

e Correlation between out-degree and average neighbor out-degree of adults and dauer (solid line: linear regression fit, shaded area: 95% confidence interval). In the context of network assortativity, this metric shows the tendency of nodes to establish connections with other nodes that exhibit a similar degree. The results presented herein are based on the out-degree of both the source and target nodes. (r = Pearson correlation coefficient, rounded to the second decimal place).

f Hub neurons in adults and dauer chemical network based on node out-degree. Horizontal black line indicates the threshold degree(Method). Hub neurons are marked in red and shaded.

Minor comments:

1. Line 240-242: “The fact that they have fewer common connections ...” It is unclear what the authors meant here. Please elaborate.

- **Response:** Thank you for pointing this out. Because we found it difficult to follow, we decided to delete the sentence to simplify the explanation.

Revised text: (line 256) **To strengthen our argument for rewiring of motor neurons in terms of target variation**, we examined “connection similarity (CS)”, which measures the similarity of connection targets of neurons between two stages (Methods).

2. Line 262: “As the last piece of the puzzle, ...” \diamond “To solve the last piece of the puzzle, ...”

- **Response:** Thank you for pointing this out.

Original text: As the last piece of the puzzle, we have reconstructed the first dauer nerve ring connectome from EM images, consisting of dense volumetric reconstructions of both neuronal and muscle cells, and their connectivity acquired using deep learning.

Revised text: (line 276) **To solve** the last piece of the puzzle, we have reconstructed the first dauer nerve ring connectome from EM images, consisting of dense volumetric reconstructions of both neuronal and muscle cells, and their connectivity acquired using deep learning.

3. Line 308-309: “However, our classification of interneurons are different from interneurons defined in 5 as it also includes modulatory neurons, which have many variable connections.” What does “it” in the sentence refers to? The authors’ classification or the classification of Witvliet et al. 2021? According to English grammar, it refers to the latter. But based on the context, it looks like the former. Please clarify.

- **Response:** Thank you for clarifying this. Later was right – classification of Witvliet et al., 2021. To avoid the confusion, we re-wrote that sentence as follows:

Original text: However, our classification of interneurons ⁴ are different from interneurons defined in ⁵ as it also includes modulatory neurons, which have many variable connections ⁵

Revised text: (line 329-330) **Neurons classified as interneurons by our definition⁴ included modulatory neurons defined by Witvliet et al. ⁵, which** have many variable connections.

4. Figure 6h. This plot does not deliver much information and lacks explanation in the legend. I fail to see how those bold arrows in the S network represent the concept of “rewiring” in the sensory subnetwork. Moreover, what do the curved arrows pointing from S to I and M mean? I suggest removing this panel or replacing it with a plot that displays more detailed information.

- **Response:** Thank you for your comments. We added additional explanation to the figure legend.

Revised text: **Fig. 6h** Summary diagram illustrating differences between dauer and adult networks. **Black curved arrows represent the remodeling of sensory outputs. Red arrows indicate increases in clustering.**

5. Line 554-559. Use past tense instead of present perfect tense.

- **Response:** Thank you for pointing this out.

Original text: Two areas of the volume have been selected from the EM volume and 10 sections from each area have been chosen to be annotated for the ground truth labels. Human annotators painted presynaptic active zones, which are darkly stained pixels near the membrane, using VAST ³² and the results have been reviewed by another annotator. Ground truth labels covering an area of 9053 μm^2 have been generated. The labels have been divided into train (7340 μm^2), validation (874 μm^2), and test (839 μm^2) datasets.

Revised text: (line 581-585)Two areas of the volume **were** selected from the EM volume and 10 sections from each area **were** chosen to be annotated for the ground truth labels. Human annotators

painted presynaptic active zones, which **were** darkly stained pixels near the membrane, using VAST³⁴ and the results **were** reviewed by another annotator. Ground truth labels covering an area of 9053 μm^2 **were** generated. The labels **were** divided into train (7340 μm^2), validation (874 μm^2), and test (839 μm^2) datasets.

6. This study uses three adult datasets. However, in some plots, e.g., Fig 1 d-e middle, Fig 2d, Fig 3c & f, Fig 4a, and Fig 6d & g, it looks like only a single adult is shown or compared. Please specify which adult dataset is used. Or if some plots are based on the average from multiple datasets, please also specify.

- **Response:** Thank you for pointing this out. We used adult-2 dataset as the representative adult, but we forgot to write down that explanation. Now, all figure legends were specified as follows:

ex) **Fig.3c** Wiring diagram of neurons shown in **a** and **b** in **adult-2** stage (left) and dauer stage (right).

In figure 6, as you commented, some plots are using different 'Adult', so it is described in figure legend.

Fig.6. a, b, c Values in adult are computed by averaging over three adult datasets. **e, f** Values are computed by averaging over all pairs of datasets.

REVIEWERS' COMMENTS

Reviewer #1 (Remarks to the Author):

The authors have satisfactorily addressed the points I raised.

Reviewer #3 (Remarks to the Author):

The authors have addressed all my comments. They have also performed the requested experiments and analyses. I do not have further comments and would like to congratulate them on their excellent work. I believe this work will make an important contribution to the field and will be highly cited. In conclusion, I recommend the publication of this manuscript in Nature Communications.

RESPONSES TO REVIEWER COMMENTS

Reviewer #1 (Remarks to the Author):

General comments

The authors have satisfactorily addressed the points I raised.

- Response: Thank you for the comments.

Reviewer #3 (Remarks to the Author):

General comments

The authors have addressed all my comments. They have also performed the requested experiments and analyses. I do not have further comments and would like to congratulate them on their excellent work. I believe this work will make an important contribution to the field and will be highly cited. In conclusion, I recommend the publication of this manuscript in Nature Communications.

- Response: Thank you for the comments.